



# Automatic controller tuning using a zeroth-order optimization algorithm

Daniel S. Zalkind[1], Emiliano Dall'Anese[1], and Lucy Y. Pao[1]

[1]Department of Electrical, Computer & Energy Engineering, University of Colorado Boulder, Boulder, CO 80309, USA

**Correspondence:** Daniel S. Zalkind (dan.zalkind@gmail.com)

**Abstract.** We develop an automated controller tuning procedure for wind turbines that uses the results of nonlinear, aeroelastic simulations to arrive at an optimal solution. Using a zeroth-order optimization algorithm, simulations using controllers with randomly generated parameters are used to estimate the gradient and converge to an optimal set of those parameters. We use kriging to visualize the design space and estimate the uncertainty, providing a level of confidence in the result.

The procedure is applied to three problems in wind turbine control. First, the below-rated torque control is optimized for power capture. Next, the parameters of a proportional-integral blade pitch controller are optimized to minimize structural loads with a constraint on the maximum generator speed; the procedure is tested on rotors from 40 to 400 m in diameter and compared with the results of a grid search optimization. Finally, we present an algorithm that uses a series of parameter optimizations to tune the lookup table for the minimum pitch setting of the above-rated pitch controller, considering peak loads

and power capture. Using experience gained from the applications, we present a generalized design procedure and guidelines for implementing similar automated controller tuning tasks.

## 1 Introduction

In this article, we present a data-driven, simulation-based optimization procedure for tuning wind turbine controllers using measures that are directly related to component design. Controller tuning influences the power capture and structural loading

on wind turbines, which are directly related to the cost of the wind energy generated. At the same time, different turbine models require different control parameters. As rotor designs are iterated upon and also customized, e.g., with larger towers, tip extensions, or for site-specific turbulence, an updated (and ideally optimized) controller is required for component design and cost specification. Given the aeroelastic turbine model, the algorithm presented in this article automatically finds the optimized parameters of the predefined control architecture, reducing the effort required of the control designer.

The wind turbine control tuning procedure is normally a manual process and often requires expert knowledge of the controller and turbine operation. An automated procedure could reduce the design cycle time of a manufacturer's research and development process or aid researchers in other disciplines of wind engineering that require a well-tuned controller without needing to worry about its finer details. Several control parameters are directly related to the performance of the turbine and must be tuned for each design iteration or model update. The simplest method to tune a controller using simulation information

is to exhaustively search the design space and then make an educated design choice of the parameter.



A systematic, simulation-based parameter search of the pitch control gains for generator speed control was first published in Hand and Balas (2000). On a single turbine, turbulent simulations were used to sample and visualize the design space against competing design measures: generator speed regulation versus blade pitch actuation. A similar data processing flow was used in Hansen et al. (2005) while the problem was formulated in a numerical optimization framework for structural load reduction;

the authors concluded that a good initial guess was only marginally worse than the optimized result and that the effort required to set up the optimization procedure was not profitable for the benefit in structural loading. Shortly thereafter, an adaptive control framework was found to be beneficial for reconciling plant-model mismatches in field testing, especially for control parameters that affect power production (Johnson et al., 2006), where even small benefits are profitable to the operator.

As the wind industry has matured and computational cost has decreased, wind turbine design increasingly relies on simu-
lation of power capture and structural loads for design analysis. As a result, system engineering tools for wind turbine design have been developed and refined, leading to updated efforts in automated controller development, with the aim of deploying tuning methods for many different turbines. One approach is to use a model-based control scheme in order to limit the control tuning effort (Bottasso et al., 2012); these authors also proposed the use of a multi-objective optimization because the value of power capture versus load reduction can vary over the turbine's lifetime. A scalar cost function was presented in Tibaldi
et al. (2014) using measures that are directly related to wind turbine component design, like peak and fatigue loads. The cost function included terms for each turbine part, with factors to capture its relative cost to the turbine.

Using measures directly related to the component design, like fatigue and extreme loads in turbulent simulations, is ideal because it most accurately reflects the eventual component design, but these simulations require more detailed and computationally expensive methods to generate the measures. Simulation-based optimization has been used to solve these problems,
where solving for the value of the cost function is expensive compared to the optimization procedure. One approach to solving these types of problems is using Response Surface Methodology (RSM) (Fu, 2014). RSM was originally developed for experimental design (Box and Wilson, 1951), but has increasingly been used with simulation information; it works by fitting a cost function to the simulation results, finding the local gradient of the fit, and optimizing the fitted cost function. The question of how to sample the parameter space remains an open question. Samples can be generated using a grid search or random
sampling. An example in the wind energy community by Moustakis et al. (2019) samples the parameter space based on a cost function that considers both where the cost is expected to be optimal and also where it is unknown. Our approach to sampling the parameter space is based on stochastic approximation or "zeroth-order optimization," which uses the sampled cost function to estimate the local gradient and then optimizes the function with proven convergence results (Ghadimi and Lan, 2013). In one of the original stochastic approximations methods, Kiefer and Wolfowitz (1952), each dimension of the decision variable
is perturbed and a finite difference method is used to approximate the gradient. Multi-point methods were developed for higher dimensional cases, where the decision variable can be perturbed in a "direction" containing multiple dimensions and the directional derivative is used to estimate the gradient (Duchi et al., 2015). If multiple directions are randomly sampled using a normal Gaussian distribution and then averaged to find the directional derivative, it is known as Gaussian smoothing, which has been shown to improve convergence rates (Hajinezhad et al., 2017).





We use a Gaussian smoothing approach to generate samples, estimate the gradient, and identify a (possibly local) minimum point. Then we use the samples to visualize the design space and provide a level of confidence in the result. Previous work in controller optimization usually only provides the cost function and goals of the optimization, whereas this work explicitly details the method for determining the sample simulations and how their results are used to iterate on control designs.

Instead of using a single cost function that attempts to account for all aspects of wind turbine design, our work solves specific wind turbine control problems that are directly related to the cost of energy. First, the optimization procedure is demonstrated on below-rated torque control to increase power capture. Next, the pitch control parameters for above-rated pitch control are optimized to reduce fatigue or extreme loads on the tower or blades with a maximum generator speed constraint. Finally, the minimum pitch setting of the pitch controller is optimized in a series of parameter optimizations aimed at reducing peak blade

loads.

This article is organized as follows. The optimization algorithm and visualization method are presented in Section 2. Applications of the algorithm for wind turbine control tuning are presented in Section 3, followed by a generalization of the design procedure and guidelines for parameter selection in Section 4. Conclusions are presented in Section 5 and the generalized wind turbine controller that is tuned in Section 3 is described in Appendix A.

## 1.1 Mathematical Notation

Superscript notation will be used to index the stage $r$ of the zeroth-order optimization algorithm: e.g., $z^r$. Additionally, $F^T$ is the transpose of $F$. If the power of any value is computed, the base will appear in parentheses: e.g., $(\gamma)^r$.

## 2 Method: Zeroth-Order Optimization

The zeroth-order optimization algorithm uses $J$ random samples near the current iteration to estimate a gradient. Then, a

typical first-order method ensues: using the estimated gradient, the descent direction and step size are chosen to produce the next iterate. The process is repeated for a number of stages $N_{\text{stage}}$ until convergence is observed. Using the cost function samples, an estimation of the design space is generated to visually verify the results. In each control tuning example presented in this article, the following unconstrained optimization problem is solved:

$$\min_{z \in \mathcal{X}} \quad \mathcal{C}(z), \tag{1}$$

where $z \in \mathcal{X} \subset \mathbb{R}^M$ is the $M$-dimensional parameter or vector of control parameters, constrained to be in a set $\mathcal{X}$ that is convex and compact; $\mathcal{C} : \mathbb{R}^M \to \mathbb{R}$ is the cost function, which we assume is differentiable, bounded from below, and its gradient is Lipschitz (Ghadimi and Lan, 2013). However, we only have access to the cost function via samples of potentially noisy simulation results.



## 2.1 Generating Samples

The algorithm begins with an initial guess $z^1$. During each stage $r = 1, 2, \ldots, N_{\text{stage}}$, the gradient is estimated by randomly sampling the design space. During each stage, sample directions $\phi_j^r \in \mathbb{R}^M$, $j = 1, 2, \cdots, J$ are parameters drawn from a random distribution. In the most general case, a standard normal distribution is used for each dimension and the vectors are normalized

to have a unit magnitude; this results in a uniformly random distribution of directions in $M$-dimensions (Hajinezhad et al., 2017). In this article, we focus on 1- and 2-dimensional parameter optimizations and will make changes to the generation of the sample directions $\phi_j^r$ to ensure an even distribution for a small number of samples.

A search sample $z_j^r$ in stage $r$ is generated according to

$$z_j^r = z^r + \mu \phi_j^r, \tag{2}$$

where $\mu$ is a smoothing parameter that determines the amount of space over which the parameter space is searched. A large value for $\mu$ helps to estimate the value of the cost function over a larger area (Section 2.6), but smaller values of $\mu$ tend to result in more accurate convergence.

The number of stages and samples-per-stage must also be chosen by the designer. A large number of samples-per-stage gives the best estimate for the gradient, but requires more simulations. During the development of this work, it was found

that a smaller number of samples-per-stage and more stages resulted in better convergence using the same total number of simulations (e.g., in Section 3.2.2).

## 2.2 Gradient Estimation

At each stage $r$, the cost $\mathcal{C}(z)$ is computed via simulation at each search sample and used to estimate the gradient

$$\bar{G}(z^r) = \frac{1}{J} \sum_{j=1}^{J} \frac{\mathcal{C}(z^r + \mu \phi_j^r) - \mathcal{C}(z^r)}{\mu} \phi_j^r. \tag{3}$$

Note that (3) differs from a finite difference method of estimating the gradient, where the factor $\phi_j^r$ would be in the denominator. Because there is uncertainty expected in the computed cost, small perturbations ($\mu \phi_j^r$) and a non-smooth cost function $\mathcal{C}(z)$ could result in noisy gradients. The gradient estimator in (3) is referred to as the random directions gradient estimator (Fu, 2014, p. 110), and maintains the convergence criterion when used in a first-order algorithm (Hajinezhad et al., 2017).

## 2.3 Determine Descent Direction

From the estimated gradient, the possible descent direction is computed:

$$d_r = -D \bar{G}(z^r), \tag{4}$$

where a diagonal matrix $D$ of positive scalars is used to relatively increase $d_r$ in the directions where the sensitivities of the cost function to parameter changes are smallest, providing a diagonal approximation to Newton's method and improving





convergence rates (Bertsekas, 1999), which leads to the following stage gain:

$$z^{r+1} = \text{Proj}_{(1-\rho)\mathcal{X}}\{z^r + d_r\alpha\}, \tag{5}$$

where $\alpha$ is the step size and $\text{Proj}_{\mathcal{X}}\{y\} := \arg\min_{x \in \mathcal{X}} \|x - y\|_2^2$ finds the closest point within the parameter bounds $\mathcal{X}$, offset with $\rho = \mu$ so that search samples in the next stage can be generated within the parameter bounds. The algorithm described

in (2)–(5) is proven to converge to a ball centered around an optimal solution (Hajinezhad et al., 2017). Next, we describe two adjustments to the original algorithm that improve performance when used in the control tuning applications presented in this article.

## 2.4 Adjustment 1: Decreasing Step Size and Line Search

A decreasing step size rule ensures convergence and a line search is used to so that the cost function does not increase in

successive iterations. After choosing a base step size $\alpha_0$, the cost of test samples

$$z^r_{a,k} = \text{Proj}_{(1-\rho)\mathcal{X}}\{z^r + d_r\alpha_k\} \tag{6}$$

are evaluated (through simulation) along the descent direction, where the step size

$$\alpha_k = \alpha_0(\beta)^{k-1} \tag{7}$$

decreases for a number of iterations $k \le k_{\max}$. An upper limit $k_{\max}$ on the number of step size samples is chosen to cap the

number of simulations that may be performed along directions that could increase the cost function. To ensure that the cost function is non-increasing during each iteration, the Armijo rule for step size is used (Boyd and Vandenberghe, 2004):

$$\mathcal{C}(z^r) - \mathcal{C}(z^r_{a,k}) > -\sigma\alpha_k\bar{G}^T d_r > 0, \tag{8}$$

where, for all examples in the following section, $\sigma = 0.05$ is chosen, a conservative value that only requires a small decrease in the cost function.

## 20 2.5 Adjustment 2: Resetting Parameter Update

Once an adequate step size is found, the parameter $z$ is updated using

$$z^{r+1} = \text{Proj}_{(1-\rho)\mathcal{X}}\{z^r + d_r\alpha_k\}, \tag{9}$$

which is the $z^r_{a,k}$ in (6) with the first $k$ that satisfies (8); since this value has already been computed, the simulation for determining $\mathcal{C}(z^r)$ for $r > 1$ does not need to be performed.

If the maximum number of step size simulations ($k = k_{\max}$) are performed and the step size rule in (8) is not satisfied, the next iteration of the parameter $z$ is chosen as

$$z^{r+1} = \arg\min_{z \in \mathcal{X}} \mathbf{C}(r), \tag{10}$$



where $\mathbf{C}(r)$ is the enumeration of the cost function at all stage samples $z^{\mathbf{r}}$, all search samples $z_{\mathbf{j}}^{\mathbf{r}}$, and all step size samples $z_{a,\mathbf{k}}^{\mathbf{r}}$ within the parameter bounds defined by $\mathcal{X}$, up until the current stage $r$:

$$\mathbf{C}(r) = \{\mathcal{C}(z^{\mathbf{r}}), \mathcal{C}(z_{\mathbf{j}}^{\mathbf{r}}), \mathcal{C}(z_{a,\mathbf{k}}^{\mathbf{r}})\}, \tag{11}$$

where $\mathbf{C}(r) \in \mathbb{R}^{n_{\mathrm{samp}}}$ has $n_{\mathrm{samp}} \leq N_{\mathrm{stage}} \times J \times k_{\mathrm{max}}$ elements, $\mathbf{r} = \{1, \ldots, r\}$, $\mathbf{j} = \{1, \ldots, J\}$, and $\mathbf{k} = \{1, \ldots, k_{\mathrm{max}}\}$. As before,
with the step size sample, since the value of the cost at this point has already been computed, it is unnecessary to compute it again for $r > 1$. Since the resetting parameter update in (10) results in a sequence of $\mathcal{C}(z^r)$ that is non-increasing, the convergence properties of the original algorithm are maintained; the same argument applies to Adjustment 1 in Section 2.4. The solution $z^{\mathrm{soln}}$ of the zeroth-order optimization is determined by the updated parameter of the final stage:

$$z^{\mathrm{soln}} = z^{N_{\mathrm{stage}}+1}. \tag{12}$$

## 2.6 Visualization

To provide confidence in the result of the zeroth-order optimization, we visualize the cost, and the measures associated with it, over the parameter space. If the minimum of the zeroth-order parameter optimization matches that of the visualization, the user can be confident in the result. The visualization method also provides a quantitative measure of the uncertainty of the estimated cost over the parameter space.

To estimate the cost and its variance over the parameter space, we use ordinary kriging. Kriging was originally developed for mining applications, where sparsely sampled information over a geographical space was used to estimate the quantity over the whole area. More recent applications of kriging include engineering design and computer experiments.

Kriging, or Gaussian process regression, is a method of interpolation that incorporates uncertainty in the area between samples. Using all the observed data from the zeroth-order parameter search at stage $r$, $\mathbf{C}(r)$ from (11), ensuring there are no repeated values, the estimated cost at $z$ is

$$\hat{\mathcal{C}}(z) = f^T(z)\hat{\beta} + \bar{\psi}^T(z)\Psi^{-1}(\mathbf{C}(r) - F\hat{\beta}), \tag{13}$$

where the first term in (13) is the generalized least squares estimate

$$\hat{\beta} = (F^T\Psi^{-1}F)^{-1}F^T\Psi^{-1}\mathbf{C}(r). \tag{14}$$

Since we are using ordinary kriging, which assumes a constant mean across the parameter space, the regression basis function

$$f(z) = 1 \quad \text{and} \quad F = f(Z) = \mathbf{1} \in \mathbb{R}^{n_{\mathrm{samp}}}, \tag{15}$$

where $Z$ is the enumeration of all sample points like in (11). The correlation matrix $\Psi$ represents the influence that nearby samples have on each other; it has the form

$$\Psi = \begin{bmatrix} \psi(z_1, z_1) & \psi(z_1, z_2) & \cdots & \psi(z_1, z_{n_{\mathrm{samp}}}) \\ \psi(z_2, z_1) & \psi(z_2, z_2) & \cdots & \psi(z_2, z_{n_{\mathrm{samp}}}) \\ \vdots & \vdots & \ddots & \vdots \\ \psi(z_{n_{\mathrm{samp}}}, z_1) & \psi(z_{n_{\mathrm{samp}}}, z_2) & \cdots & \psi(z_{n_{\mathrm{samp}}}, z_{n_{\mathrm{samp}}}) \end{bmatrix} \tag{16}$$





and is made up of scalar Gaussian correlation functions

$$\psi(z_1, z_2) = \exp\left(-\sum_{i=1}^{M}(|z_{1,i} - z_{2,i}|/\nu_i)^2\right), \tag{17}$$

where $\nu_i$ is the distance at which the influence is $e^{-1}$ or 37% in the $i^{\text{th}}$ dimension (Martin and Simpson, 2008). The second term in (13) interpolates or "pulls" the estimate towards the observed values using the correlation vector

$$\bar{\psi}^T(z) = \begin{bmatrix} \psi(z, z_1) & \psi(z, z_2) & \cdots & \psi(z, z_{n_{\text{samp}}}) \end{bmatrix}. \tag{18}$$

The mean squared error, or variance, of the cost at $z$ is determined by

$$\text{MSE}[\hat{\mathcal{C}}(z)] = \sigma_{\text{proc}}^2\left(1 - \begin{bmatrix} f(z) & \bar{\psi}^T(z) \end{bmatrix} \begin{bmatrix} 0 & F^T \\ F & \Psi \end{bmatrix} \begin{bmatrix} f(z) \\ \bar{\psi}(z) \end{bmatrix}\right), \tag{19}$$

where

$$\sigma_{\text{proc}}^2 = \frac{1}{n_{\text{samp}}}\left(\mathbf{C}(r) - F\hat{\beta}\right)^T \Psi\left(\mathbf{C}(r) - F\hat{\beta}\right) \tag{20}$$

is the process variance. As the unobserved point $z$ moves away from the observed samples, the second term in (19) approaches zero and the variance approaches $\sigma_{\text{proc}}^2$.

The correlation function parameters $\nu_i$ are estimated using a maximum likelihood estimator to be consistent with the observed data. To perform this optimization, we use the ooDACE toolbox to fit the correlation function and kriging model (Couckuyt et al., 2013). Problems can arise when using kriging for simulation-based optimization because of ill-conditioned correlation matrices (Booker et al., 1999). When samples cluster near the optimal solution, closely spaced samples with different

values can result in very small values of $\nu_i$ and ill-conditioned correlation matrices. One solution is to add a constant to the diagonal of $\Psi$ (Sasena, 2002). We implement this using "stochastic kriging", where the samples are assumed to have uncertainty and is equivalent adding their variance to the diagonal of $\Psi$ (Couckuyt et al., 2013). Additionally, the lower and upper bounds on the values of $\nu_i$ depend on the minimum and maximum spacing of the distance between samples, respectively (Martin and

Simpson, 2008).

## 2.7  Settling Function

To measure the number of stages the optimization procedure requires to find the minimum of the cost function, we define the settling function

$$s(r) = \frac{\mathcal{C}(z^{\text{soln}}) - \mathcal{C}(z^r)}{\mathcal{C}(z^{\text{soln}}) - \mathcal{C}(z^1)}, \tag{21}$$

which is a linear transformation that represents the fraction of change in cost at each stage $\mathcal{C}(z^r)$, compared to the overall change in cost function. The initial cost $\mathcal{C}(z^1)$ is mapped to $s(1) = 1$ and the cost of the solution $\mathcal{C}(z^{\text{soln}})$ is mapped to $s(N_{\text{stages}} + 1) = 0$. Often, we perform more stages than is necessary and use this settling function to determine how many stages are required to achieve some percentage of the change in cost function.



## 3 Applications in Wind Turbine Control Tuning

In this section, we present three examples of using zeroth-order parameter optimization to tune the parameters of wind turbine controllers. As an initial demonstration, we optimize a one-dimensional parameter to maximize power capture through torque control in below-rated operation. Next, we present the motivating example for this work, a two-dimensional parameter opti-

mization for a standard pitch controller, with the goal of regulating generator speed so that loads are minimized, subject to a constraint on the maximum generator speed. Finally, we demonstrate how a series of one-dimensional parameter optimizations can be used to determine the minimum pitch setting of the pitch controller for controlling peak blade loads.

### 3.1 Optimal Torque Control Gain

In below-rated (Region II) operation, the generator torque is typically controlled using $\tau_g = k_{\text{opt}}\omega_g^2$, which controls the rotor

speed to its optimal tip speed ratio, where $\omega_g$ is the generator speed. The optimal gain $k_{\text{opt}}$ depends on a number of aerodynamic properties (Johnson et al., 2006):

$$k_{\text{opt}} = \frac{\pi \rho_{\text{air}} R^5 C_{P,\text{max}}}{2\lambda_{\text{opt}}^3 G^3}, \tag{22}$$

where $\rho_{\text{air}}$ is the air density, $R$ is the rotor radius, $C_{P,\text{max}}$ is the maximum power coefficient, $\lambda_{\text{opt}}$ is the optimal tip speed ratio, and $G$ is the gearbox ratio. We add a multiplicative factor to account for uncertainties in the aerodynamic properties and to

allow the gain to be increased or decreased, resulting in the control law

$$\tau_g = k_{\text{fact}} k_{\text{opt}} \omega_g^2. \tag{23}$$

In practice, a value other than $k_{\text{fact}} = 1$ is found to be optimal for a realistic turbulent wind input.

The goal of this optimization procedure is to find the gain $k_{\text{fact}}$ that results in the greatest energy capture. To maintain the form of a minimization problem, we solve

$$\min_{z=k_{\text{fact}}} -\bar{P}(z), \tag{24}$$

where the cost function $\mathcal{C}(z) = -\bar{P}(z)$ is the negative of the mean generator power, and the optimization parameter $z = k_{\text{fact}}$.

At each stage, $J = 2$ samples are simulated to compute the cost function and estimate the gradient. In this one-dimensional problem, no dimensional scaling is required, thus $D = 1$. With $J = 2$, we set the search direction to

$$\phi_j^r = \{-1, 1\} \quad \text{for} \quad j = 1, 2. \tag{25}$$

The optimal value of $k_{\text{fact}}$ is expected to be between 0.3 and 1.7, so these values are set as hard bounds. A search range of $\mu = 0.05$ is set to adequately search the space and estimate meaningful gradients. For tuning controllers of turbines with different power ratings, the base step size is scaled with the inverse of the initial simulation's average power $\bar{P}(z^1)$. Larger power values result in larger gradients, since the scale of the parameter is constant for all rotors (it should ideally be 1), the step

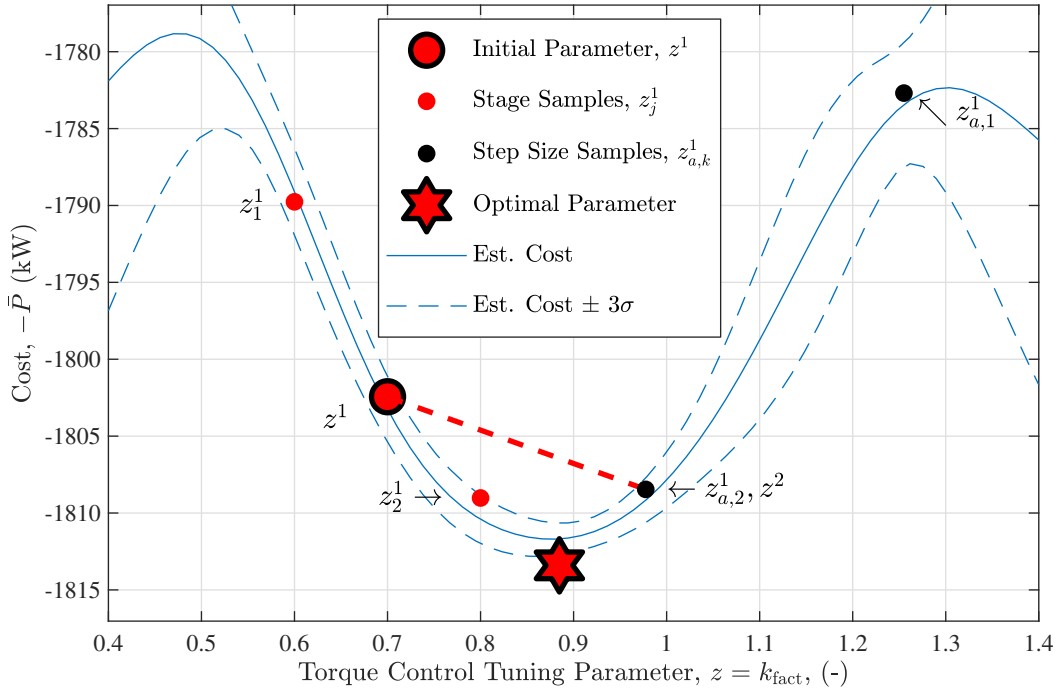

**Figure 1.** The first iteration of the one-dimensional parameter tuning for the optimal torque control gain. Starting with the initial parameter $z^1$, random samples $z_1^1$ and $z_2^1$ are generated to estimated the gradient. Note that $\mu = 0.1$ in this figure for clarity. Test samples $z_{a,1}^1$ and $z_{a,2}^1$ are evaluated in the gradient direction until the cost decreases and the next stage parameter $z^2$ is determined. The estimated cost and uncertainty are found using (13) and (19), respectively, where $\sigma(z) = \sqrt{\mathrm{MSE}[\hat{C}(z)]}$.

size should be reduced to maintain the same rate of descent. Note that a positive step size is required, even though the cost is negative, and a maximum of three step size sample simulations are performed ($k_{\max} = 3$). A summary of the parameters used in the torque control parameter optimization are shown in Table 1 and an illustration of the first iteration is shown in Fig. 1.

The parameter optimization was performed on the NREL-5MW reference turbine with the standard lookup-table-based torque controller in Jonkman et al. (2009). The algorithm finds the optimal $k_{\mathrm{fact}}$ and converges in 7 stages (Fig. 2b). The full procedure, with 7 stages, performs 31 simulations in total, which includes step size samples and the initial guess; the average power is increased by 0.67%, compared to $k_{\mathrm{fact}} = 1$. The use of turbulent simulations contributes noise to the signal that determines the cost function, which is apparent by the non-smooth behavior of the cost samples with respect to the gain factor parameter in Fig. 2a. However, the algorithm appears to be robust to these uncertainties.

## 3.2 Pitch Control for Generator Speed Regulation

In this section, we optimize the parameters of an above-rated blade pitch controller for load reduction and generator speed regulation. Each time a new rotor is designed, the pitch controller should be tuned so that the structural loads can be computed





**Table 1.** Design choices for 1-D parameter search to optimize the torque gain in below-rated control.

|  | Parameter | Variable | Value |
|---|---|---|---|
| Stage & | Number of stages | $N_{\text{stage}}$ | 7 |
| sample size | Samples per stage | $J$ | 2 |
| | Newton's method approximation | $D$ | 1 |
| | Lower & upper bounds on search | $k_{\text{LB}}$ & $k_{\text{UB}}$ | 0.3 & 1.7 |
| | Sample search radius & smoothing parameter | $\mu$ | 0.05 |
| | Base step size | $\alpha_0$ | $10/\bar{P}(z^1)$ |
| Step | Armijo decrement factor | $\beta$ | 0.5 |
| size | Armijo threshold | $\sigma$ | 0.05 |
| | Max. step size iterations | $k_{\text{max}}$ | 3 |

to design the various hardware components of the wind turbine. As will be seen, the pitch controller affects the loads that drive turbine design. The procedure for tuning the gain-scheduled proportional-integral (PI) controller is detailed in Appendix A. First, steady-state simulations at above-rated wind speeds are used to determine the turbine operating points and aerodynamic parameters at various pitch angles, which parameterizes the gain scheduling. The final, and most involved, step is to tune the

natural frequency ($\omega_{\text{reg}}$) and damping ratio ($\zeta_{\text{reg}}$) of the "regulator mode," which represents the generator speed response to a disturbance (wind) input. The following optimization procedure aims to find an optimal set of parameters ($\omega_{\text{reg}}, \zeta_{\text{reg}}$) so that structural loads are minimized and adequate generator regulation is maintained.

In general, reducing the bandwidth of the pitch controller by choosing a lower natural frequency $\omega_{\text{reg}}$ reduces the structural loading, which we denote with $M$ in the following. However, controllers with lower natural frequencies allow greater generator

speed transients, which is acceptable up to some maximum constraint. If the generator speed exceeds some threshold $\omega_{g,\text{hard}}$, most turbines enter into a shutdown procedure to avoid further damage, which reduces the availability of the turbine and the net annual energy production; this must be avoided.

First, we reformulate the constrained optimization

$$\min_{z=(\omega_{\text{reg}},\zeta_{\text{reg}})} M(z) \tag{26}$$

$$\text{subject to} \quad \omega_g \leq \omega_{g,\text{hard}}, \tag{27}$$

as an unconstrained problem to use the algorithm described in Section 2. The cost function is augmented so that the optimization problem has the form in (1), namely

$$\mathcal{C}(z) = M(z) + B(z), \tag{28}$$

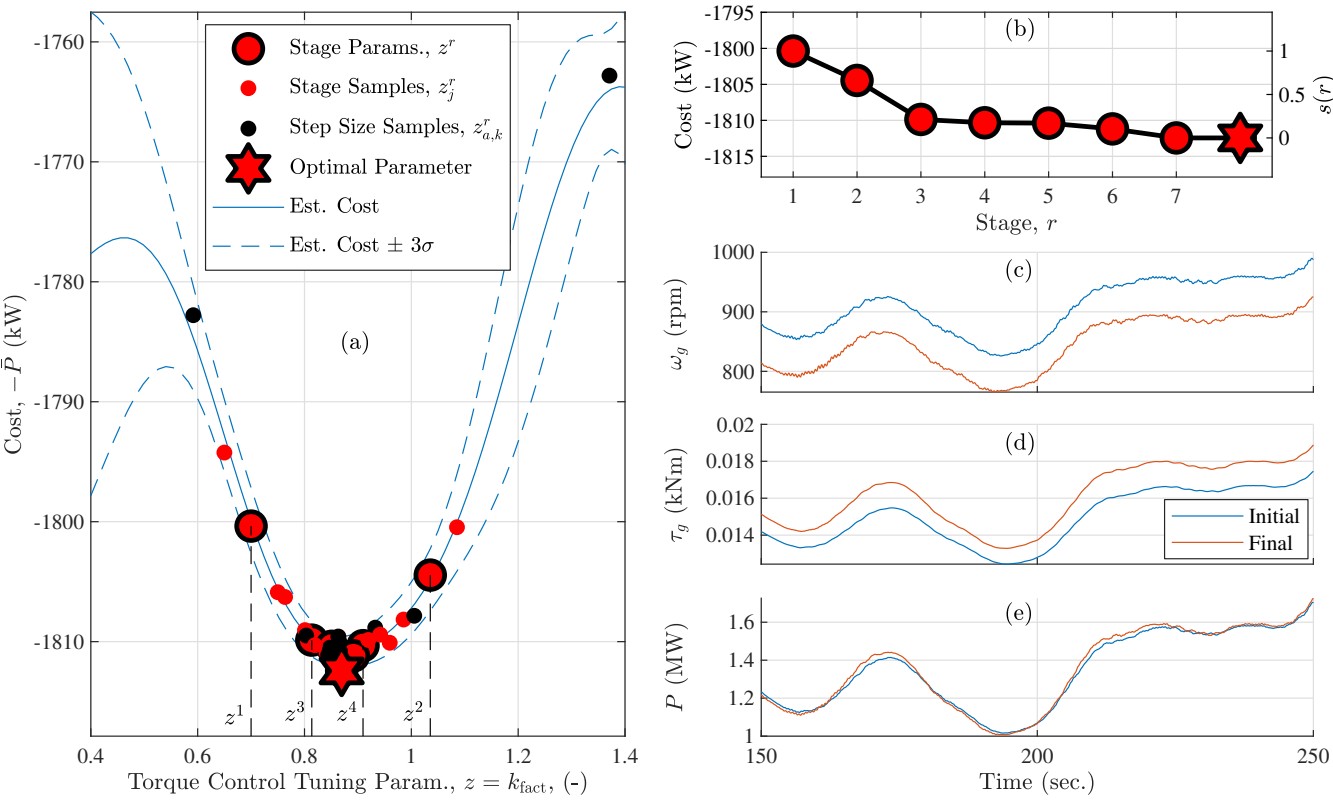

**Figure 2.** One-dimensional parameter tuning for the optimal torque control gain of the 5MW reference turbine in Class A turbulence using the negative mean generator power ($-\bar{P}$) as the cost function. In below-rated control, the generator speed ($\omega_g$) is controlled using the generator torque ($\tau_g$). The estimated cost and uncertainty are found using (13) and (19), respectively, where $\sigma(z) = \sqrt{\mathrm{MSE}[\hat{C}(z)]}$. The settling function $s(r)$ is defined in (21).

where $B(z)$ is a boundary function that penalizes samples that have a maximum generator speed that exceeds some "soft" generator speed constraint $\omega_{g,\mathrm{soft}}$,

$$
B(z) = \begin{cases} 0 & \text{if } \omega_{g,\mathrm{max}}(z) < \omega_{g,\mathrm{soft}} \\ k_B(\omega_{g,\mathrm{max}}(z) - \omega_{g,\mathrm{soft}})^2 & \text{otherwise.} \end{cases} \tag{29}
$$

A quadratic boundary function is used so that the cost is differentiable, even when an non-feasible solution is sampled. The
5    factor $k_B$ is chosen to provide a sufficient penalty on high generator speeds, but not so high that exceedingly large gradients are determined from the gradient estimation in (3), which can be problematic for the algorithm.

$$
k_B = \frac{c_{\mathrm{max}}}{(\omega_{g,\mathrm{hard}} - \omega_{g,\mathrm{soft}})^2} \tag{30}
$$

ensures that the barrier function $B(z) = c_{\mathrm{max}}$ when the maximum generator speed equals the hard generator speed constraint $\omega_{g,\mathrm{hard}}$.



To adapt the cost function to different rotors and load measures,

$$c_{\max} = \frac{1}{12} M(z^1), \tag{31}$$

where $M(z^1)$ is the load measure of the initial stage sample and the factor $\frac{1}{12}$ based on experience gained using the algorithm with simulation results. A smaller factor does not penalize maximum generator speeds enough, leading to possibly infeasible

solutions that violate (27). Factors greater than $\frac{1}{12}$ were found to create large gradients that lead the iterates away from the constraint boundary; typically, the optimal solution is found close to that boundary.

In most cases, the initial parameter set $z^1$ is chosen to be near values that were tuned manually, but offset (usually with a higher natural frequency) to allow the algorithm to converge properly. If the parameters were not previously tuned, the values suggested in the NREL-5MW reference manual (Jonkman et al., 2009) are chosen as the initial parameter set.

**Table 2.** Parameters used for the 2-D speed regulator control tuning procedure for all rotors tested. The effects of the number of stages $N_{\text{stage}}$ and samples per stage $J$ on the algorithm's performance are investigated in Section 3.2.2.

|  | Parameter | Variable | Value |
|---|---|---|---|
| | Soft constraint | $\omega_{g,\text{soft}}$ | 1300 rpm |
| Cost model | Hard constraint | $\omega_{g,\text{hard}}$ | 1400 rpm |
| | Cost at hard constraint | $c_{\max}$ | $M(z^1)/12$ |
| Stage & | Number of stages | $N_{\text{stage}}$ | 7, 12 |
| Sample size | Samples per stage | $J$ | 3, 4, and 10 |
| Newton's method approximation matrix | | $D$ | diag([0.25,1]) |
| Lower & upper parameter | | $(\omega_{\text{reg}}, \zeta_{\text{reg}})_{\text{LB}}$ | $(0.01, 0.1)$ |
| Bounds on search samples | | $(\omega_{\text{reg}}, \zeta_{\text{reg}})_{\text{UB}}$ | $(1, 3)$ |
| | Base step size | $\alpha_0$ | $3/M(z^1)$ |
| Step | Armijo decrement factor | $\beta$ | 0.5 |
| Size | Armijo threshold | $\sigma$ | 0.05 |
| | Max. step size iterations | $k_{\max}$ | 3 |
| Sample search radius & smoothing parameter | | $\mu$ | 0.05 |

A summary of the parameters used to tune all the rotors in this study is presented in Table 2. The algorithm is tested using different number of stages $N_{\text{stage}}$ and samples per stage $J$ in Section 3.2.2. The best results were achieved using a quasi-deterministic search direction,

$$\phi_j^r = [\cos\psi_j, \sin\psi_j]^T, \tag{32}$$

where

$$\psi_j = \psi_0 + \frac{2\pi j}{3}, \tag{33}$$



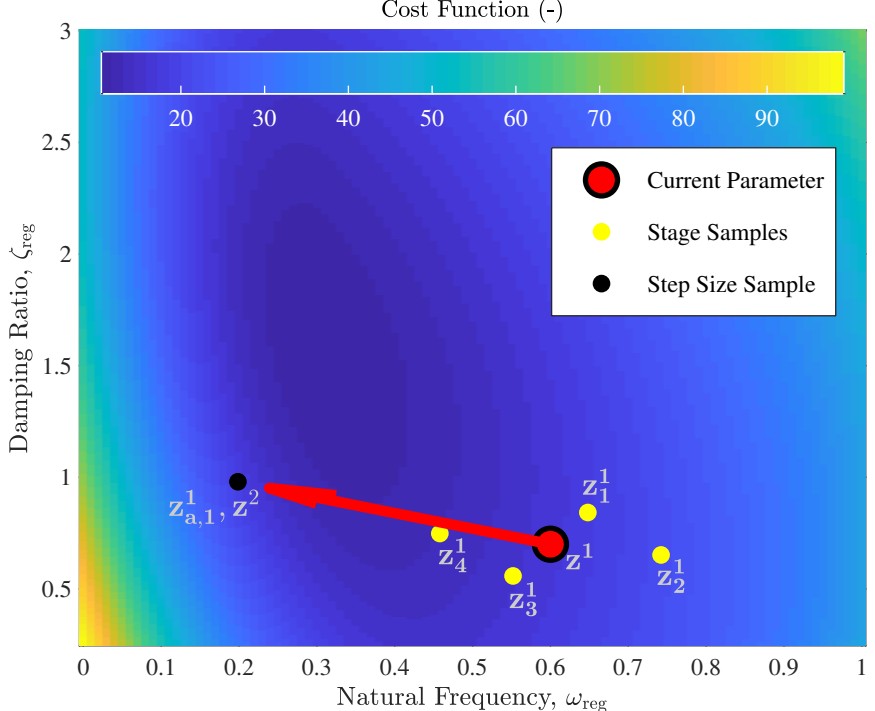

**Figure 3.** First iteration of the zeroth-order parameter optimization algorithm for the two-dimensional pitch control tuning. Random samples $z_j^1$, $j = 1...4$, are generated near the initial guess $z^1$ to estimate the gradient. Note that $\mu = 0.15$ in this figure for clarity. The sample $z_{a,k}^1$, $k = 1$ is tested along the descent direction until a sample with a decreasing cost function is found, which becomes the next guess $z^2$ for the pitch control parameter.

is used to evenly space the samples in the two dimensions $(\omega_\text{reg}, \zeta_\text{reg})$, and $\psi_0$ is randomly generated according to $\psi_0 \sim U(0, 2\pi)$, resulting in the generated samples $z_j^1$ in Fig. 3.

The cost function is more sensitive to changes in $\omega_\text{reg}$ than it is to changes in $\zeta_\text{reg}$, so $D = \text{diag}([0.25, 1])$ was chosen to relatively increase the search direction in the $\zeta_\text{reg}$ dimension. Hard bounds on $(\omega_\text{reg}, \zeta_\text{reg})$ are chosen to avoid unstable parameter 5 sets. The base step size $\alpha_0$ scales with the inverse of the initial load $M(z^1)$ so that the algorithm works for turbine models of different sizes, with initial loads specified in Table 3.

The algorithm was tested on a range of rotor models with different wind classes and load measures. First, the pitch control parameters of the NREL-5MW reference model are optimized, starting from the parameters specified by the NREL-5MW reference manual (Jonkman et al., 2009), and using the tower base moment (fore-aft) damage equivalent load (DEL) as the 10 load measure. For three different wind classes (1A, 1B, and 1C), with different turbulence levels (A–highest and C–lowest), the parameters were optimized and an example is shown in Fig. 4. In Fig. 5, the estimated cost (a), load (c), and maximum generator speed (e) across the parameter space is shown, along with the estimated uncertainty in (b), (d), and (f). The lowest turbulence level (Class 1C) has the lowest optimal natural frequency $\omega_\text{reg}$, since the reduced turbulence results in lower generator speed



**Table 3.** Summary of test cases and results from a single zeroth-order parameter tuning for speed regulation control using the parameters in Table 2 with $N_{\text{stage}} = 7$ and $J = 3$.

| Test Case | | | Start Load, $M(z^1)$ | Start Param., $z^1$ | Final Load, $M(z^{\text{opt}})$ | Final Param., $z^{\text{opt}}$ |
|---|---|---|---|---|---|---|
| Turbine | Wind Class | Load Measure | (MNm) | $(\omega_{\text{reg}}^1, \zeta_{\text{reg}}^1)$ | (MNm) | $(\omega_{\text{reg}}^{\text{opt}}, \zeta_{\text{reg}}^{\text{opt}})$ |
| | 1A | Tower | 17.5 | (0.60, 0.70) | 11.8 | (0.10, 2.08) |
| NREL-5MW | 1B | Base DEL | 16.3 | (0.60, 0.70) | 10.7 | (0.11, 1.52) |
| | 1C | (fatigue) | 15.0 | (0.60, 0.70) | 8.94 | (0.085, 1.60) |
| SUMR-13A | | | 69.5 | (0.45, 1.00) | 63.2 | (0.21, 1.11) |
| SUMR-13B | | Max. Blade | 76.4 | (0.60, 0.70) | 65.4 | (0.59, 1.59) |
| SUMR-13C | 2B | Root | 105 | (0.50, 1.25) | 65.1 | (0.18, 1.87) |
| SUMR-25 | | (extreme) | 180 | (0.25 ,0.70) | 151 | (0.06, 0.76) |
| SUMR-50 | | | 451 | (0.19, 0.47) | 423 | (0.36, 1.75) |
| SUMR-D | Custom | | 0.137 | (0.60 ,0.70) | 0.121 | (0.46, 1.46) |

transients. In each case of the NREL-5MW reference model, the optimized parameters have a lower natural frequency and higher damping ratio than the original setting.

The optimization procedure was also performed for each rotor design in the Segmented Ultralight Morphing Rotor (SUMR) project (Loth et al., 2016); for these rotors, the design driving load case for blade design was the maximum blade root bending

moment. In practice, the combined edgewise and flapwise load is used for design, but since the edgewise load is deterministic, we used the maximum flapwise load as the load measure for optimization, which is a good indicator of maximum combined loads. The SUMR rotor radii range in size from 22 m to 240 m (Table B1), and the same optimization parameters (Table 2) were used for each optimization procedure, albeit with different initial conditions and loads, which adapt the cost function and step size accordingly. The optimization procedure generally settles on a lower natural frequency and higher damping ratio than

the initial guesses (Table 3), which has the effect of producing the lowest control bandwidth (for reducing loads) but only to the point so that the generator speed constraint is not exceeded.

### 3.2.1   Discussion of Low Natural Frequency, High Damping Ratio Regulator Mode

For most of the rotors in this section, the baseline rotor speed proportional-integral (PI) control parameters are optimized to have a regulator mode with a lower natural frequency and higher damping ratio than the initial guess (Table 3). To understand

why this is the case, we must consider the cost function of the optimization. In this tuning procedure, our goal is to minimize structural loading with a constraint on the maximum generator speed. From Figs. 4(c) and (d), we see that the collective blade pitch angle $\theta_c$ has a large effect on the thrust-based structural loading; this includes tower fatigue and blade peak loads. Fig. 4 shows that, in most cases, pitch and loads mirror each other: when pitch increases, loads decrease, and vice versa. A good



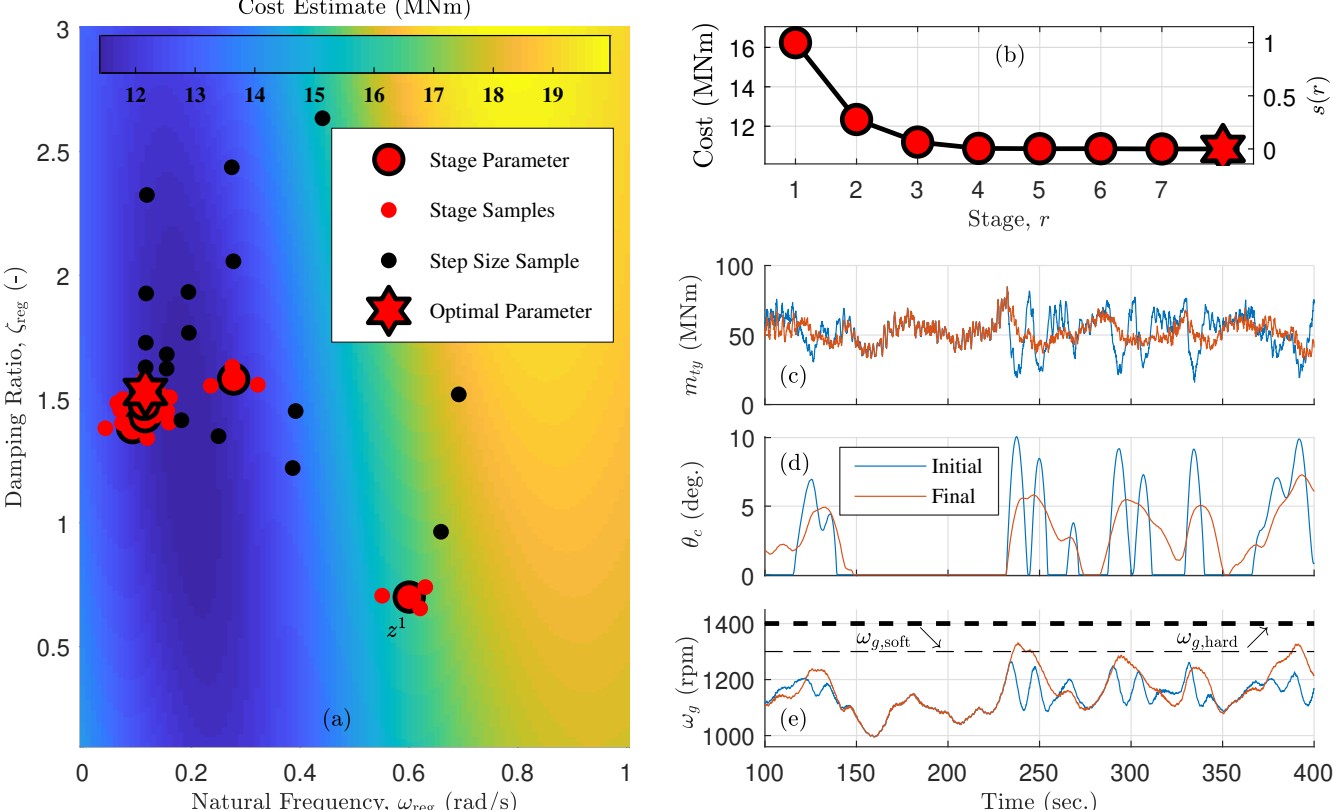

**Figure 4.** Results of using the zeroth-order optimization for tuning the pitch control regulator mode (natural frequency $\omega_{\text{reg}}$ and damping ratio $\zeta_{\text{reg}}$) of the 5MW reference turbine in Class B turbulence using the tower base fore-aft ($m_{ty}$) DEL as the load measure, which is indirectly controlled via $\theta_c$, the collective blade pitch control; $\theta_c$ is primarily responsible for regulating the generator speed $\omega_g$. The settling function $s(r)$ is defined in (21).

example occurs between 230–250 seconds in the timeseries of Fig. 4. The direct effect of the blade pitch signal on the load signal is the primary reason for the optimal PI gains (or regulator mode parameters) found in this section.

PI gains derived from a regulator mode with a low natural frequency result in less pitch actuation, thus less change in the load. Higher natural frequencies result in faster and more frequent pitch control variations, which translate to the structural
5  load signals and increase fatigue loading. A controller with a high natural frequency can also be problematic when the wind speed decreases. Because the underlying controller is trying to regulate the generator speed, the pitch will decrease during a wind lull to maintain the generator speed at its rated value, which can also lead to large peak loads, especially when an increase in wind speed follows.

High damping ratios are also found to be optimal when using the described cost function. A generator speed response and
10  pitch control response with a high damping ratio lacks any overshoot and secondary transients when the system is subjected

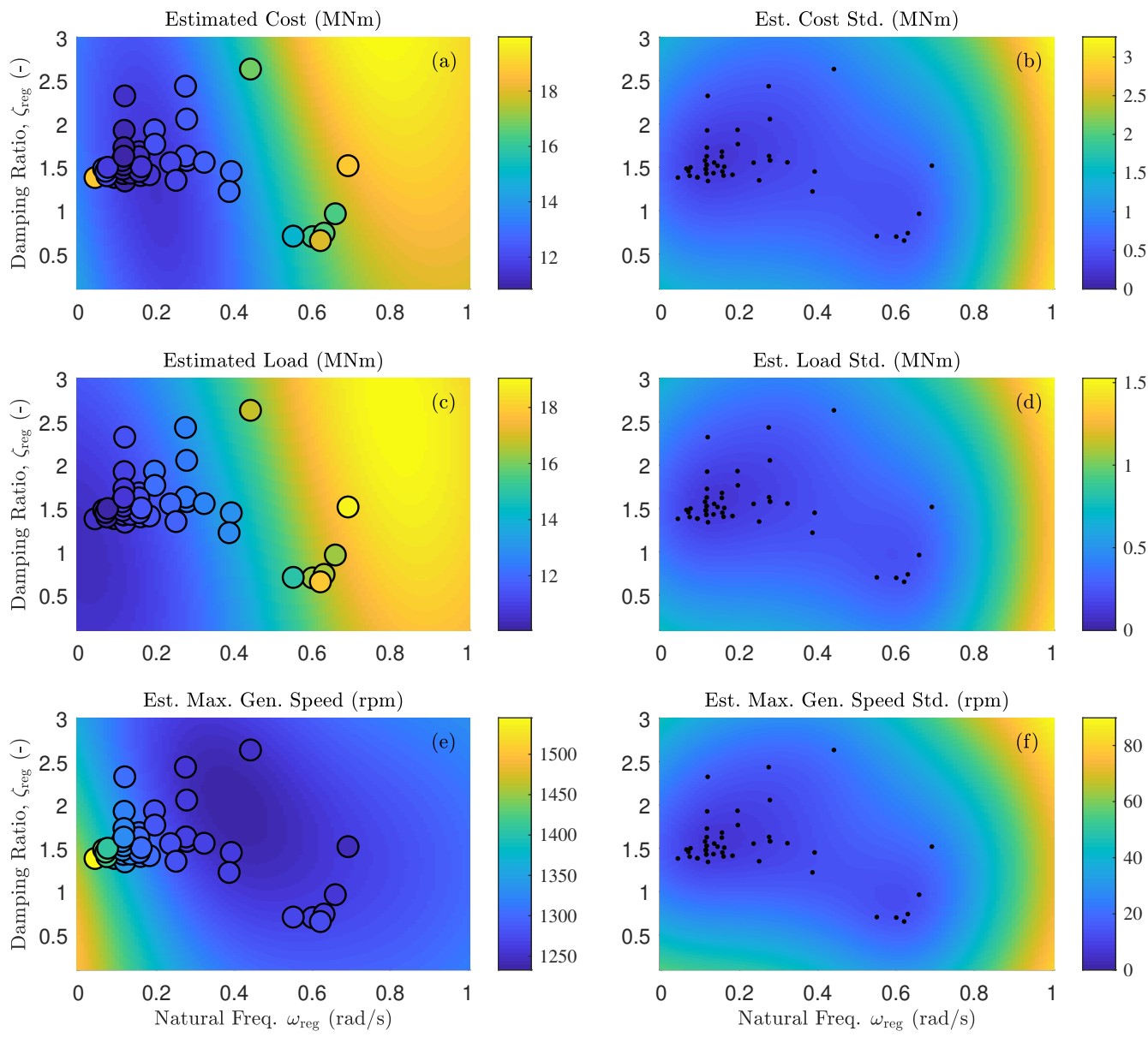

**Figure 5.** Cost (a), load (c), and generator speed (e) estimated values and standard deviations (std.; b, d, and f) using the kriging visualization described in Section 2.6. The sample values are shown in (a), (c), and (e) and their locations are depicted in (b), (d), and (f).





to a disturbance (wind). Secondary transients and overshoot in the pitch command result in load transients. The original NREL-5MW controller (where $\omega_{\text{reg}} = 0.6$, $\zeta_{\text{reg}} = 0.7$) has regulator mode poles at $-0.35 \pm j0.45$, indicative of a fast response with overshoot and transients in the pitch and generator speed signals. The optimized controller in Class 1A turbulence (with $\omega_{\text{reg}} = 0.10$, $\zeta_{\text{reg}} = 2.08$) has two real poles at $-0.40$ and $-0.025$, which results in a fast initial pitch response and a slower secondary response.

When comparing the PI-gains of the original versus optimized controller, we see that the proportional gains are of similar magnitudes, but the integral gain is much less in the optimized set of gains. The original NREL-5MW gains are $k_{P,0} = 2.3 \times 10^{-2}$ s and $k_{I,0} = 1.0 \times 10^{-2}$, whereas the optimized gains (in Class 1A turbulence) are $k_{P,0} = 1.1 \times 10^{-2}$ s and $k_{I,0} = 0.023 \times 10^{-2}$. These optimized gains reflect the cost function and control goal. The optimized proportional gain is still large enough to mitigate generator speed transients, ensuring that the generator speed does not exceed the maximum threshold, while the integral gain is reduced because it causes transients in the blade pitch and structural loads. Since our primary goal is not to regulate the generator speed to some fixed set point, but instead constrain its maximum value, integral control is less important. If the cost function included a term related to regulating the generator speed to its rated set point, using e.g., mean squared error compared to the rated generator speed, the optimal integral gains might be larger. However, we believe that a controller that constrains extreme events and maximizes power capture better reflects the overall wind turbine design goals.

### 3.2.2 Results: Comparison with Grid Searching

To quantify the performance of the zeroth-order optimization (ZOO), we compare it, in terms of the number of simulations and optimal cost, with a grid search optimization for tuning the pitch controller of the NREL-5MW reference turbine in Class 1A turbulence. The same area spanned by the hard bounds of the zeroth-order method (Table 2) is sampled by a $N_{\text{grid}} \times N_{\text{grid}}$ grid, with $N_{\text{grid}} = 6$, 8, and 10. The cost, defined by (28)–(31) and sampled using the $N_{\text{grid}} = 10$ grid search is shown in the background of Figs. 6(a)–(e). The parameter $z$ with the minimum cost over all simulations in the search is the optimal parameter $z^{\text{opt}}$.

The ZOO procedure outlined in Section 2 is performed three times for each of the following cases. Each procedure uses randomly generated samples that should result in different optimal parameters for each instance. We use four different initial conditions, distributed so that one is in each of the four quadrants spanning the bounded parameter space. The starting location in each quadrant was generated randomly, except for the bottom, right quadrant in Fig. 6(d), which is the suggested parameter set, $z^{\text{sug}} = (\omega_{\text{reg}}, \zeta_{\text{reg}}) = (0.6, 0.7)$, defined in the NREL-5MW reference manual (Jonkman et al., 2009). The ZOO was performed with $N_{\text{stage}} = 7$ stages using $J = 3$, 4, and 10 samples-per-stage and also with $N_{\text{stage}} = 12$ using $J = 3$ samples-per-stage. Theoretical results suggest that better gradient estimates (from a larger number of samples-per-stage $J$) result in convergence within a ball with a smaller radius centered around an optimal solution (Hajinezhad et al., 2017). The optimal parameter set $z^{\text{opt}}$ found in each instance of the ZOO is shown in Figs. 6(a)–(d). We compare the cost of the ZOO method with the grid search optimization in terms of the defined cost function in (28)–(31) (Fig. 6). The results are normalized to the cost function found using the suggested parameter set $z^{\text{sug}}$.





**Figure 6.** Figs. (a)–(d) show the result of performing the zeroth-order optimization (ZOO) for the baseline rotor speed controller using $N_{\text{stage}} = 7$ and 12 stages and $J = 3$, 4, and 10 samples-per-stage, using four different initial conditions ($z^1$). The background image of Figs. (a)–(e) is of the cost function, sampled using a grid search with a $10 \times 10$ level of precision using the hard bounds in Table 2 and normalized to the cost when $z^1 = z^{\text{sug}}$. Fig. (e) shows the optimal solutions of three different grid search resolutions. Fig. (f) compares the optimal costs, normalized to the initial cost when $z^1 = z^{\text{sug}}$, compared with the number of simulations used to find the result.




Compared to $z^{\text{sug}}$, all of the methods result in a 20% to 26% reduction in the cost function, with about a 1% standard deviation in the results. For fewer than 80 simulations, ZOO performs better than the grid search benchmark in almost every case. The optimal cost decreases with increasing $J$ and total number of simulations on average, but not necessarily always. However, in terms of efficiency on a per-simulation basis, $J = 3$ (blue in Fig. 6) achieves similar results to those found using $J = 10$ (yellow in Fig. 6). Additional stages ($N_{\text{stage}} = 12$) with $J = 3$ (purple in Fig. 6) decrease the cost function further; the optimal cost of this case is the best we tested in terms of the number of simulations and cost reduction. By a small margin (1–2%), using the initial parameter $z^1 = z^{\text{sug}}$ in the bottom, right quadrant in Fig. 6(d) performed better than the other $z^1$ locations shown in Figs. 6(a)–(c). In Figs. 6(a)–(d), we see that if we use a $z^1$ that is closer to the area where $z^{\text{opt}}$ is found, there is less variation in $z^{\text{opt}}$; there is also less variation the minimum cost $\mathcal{C}(z^{\text{opt}})$.

## 3.3 Minimum Pitch Setting for Peak Load Reduction

In this final example, a series of 1-dimensional parameter optimizations will be used to tune the minimum pitch setting of the above-rated pitch controller described in Appendix A and shown in Fig. A1. Increasing the minimum pitch setting can reduce the peak blade and tower loads. However, it also slightly reduces power capture. To represent this trade-off, the cost function

$$C(z) = \kappa \frac{M(z)}{M(z^0)} + (1 - \kappa) \frac{\bar{P}(z^0)}{\bar{P}(z)} \tag{34}$$

will be minimized, quantifying the relative importance $\kappa$ between reducing peak loads and reducing power capture, where $z = \theta_{\min}(u)$ is the minimum pitch setting at wind speed $u$, $M$ is the peak blade load, and $\bar{P}$ is the mean generator power of a turbulent simulation with mean wind speed $u$. A value of $\kappa = 0.01$ is used, which represents a 10% reduction in peak load being roughly equal to a 0.1% decrease in power capture; this parameter can be tuned by the control designer based on the goals of the design, however, the feasibility of the optimization problem should be verified.

During the load analysis of a control design, a number ($N_{\text{seeds}}$) of randomly generated turbulent seeds are used to simulate the turbine across wind speeds to identify peak loads on the various components. Often, peak loads on the blade and tower occur in situations where there is first a lull in the wind speed, which causes the pitch angle to decrease, followed by an increase in wind speed. If the pitch controller does not react in time, the combination of high wind speeds and low pitch angles causes a large thrust on the rotor. However, if the minimum allowable pitch is increased, the peak loads resulting from wind speed lulls can be reduced. An example is shown in Fig. 7(c) and (d) at 200 seconds. While these events are fairly common in simulation, not all produce equal peak loads; the minimum pitch setting is optimized for the worst case simulation.

A wind speed estimate, which can be found using, e.g., one of the methods in Soltani et al. (2013), is used to determine the minimum pitch setting of the above-rated pitch controller. A smooth lookup table is generated using a cubic spline interpolation and a table of three minimum blade pitch settings with breakpoints at above-rated wind speeds, in addition to a breakpoint in below-rated wind speeds and one above the cut-out wind speed. The minimum pitch setting is non-decreasing with respect to wind speed. An example is shown in Fig. 8. The minimum pitch at each above-rated breakpoint will be optimized using the zeroth-order optimization procedure previously described.



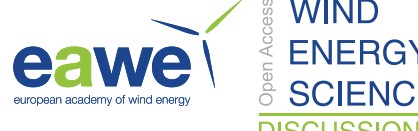

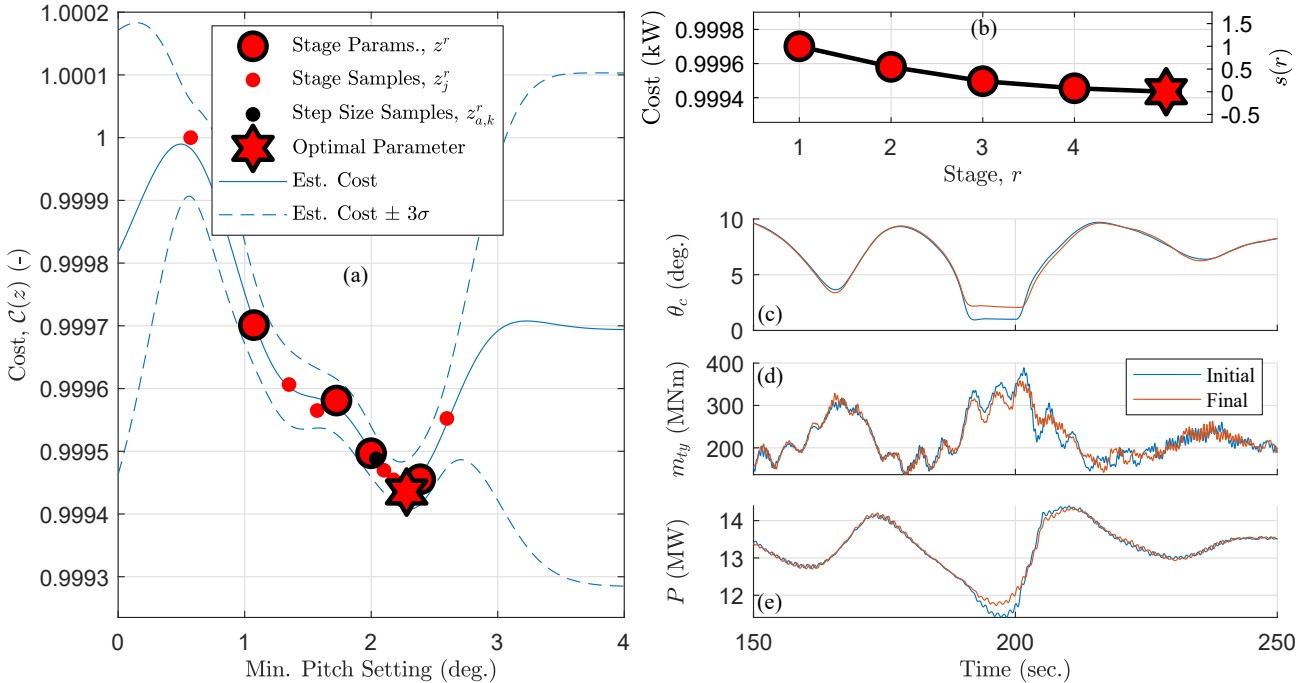

**Figure 7.** A zeroth-order optimization for the minimum pitch setting at $12\ \text{ms}^{-1}$. The cost $\mathcal{C}(z)$ in (34) is a function of the peak tower load $m_{ty}$ and mean of the generator power $P$ in the simulation, where $\theta_c$ is the collective pitch angle. The settling function $s(r)$ is defined in (21).

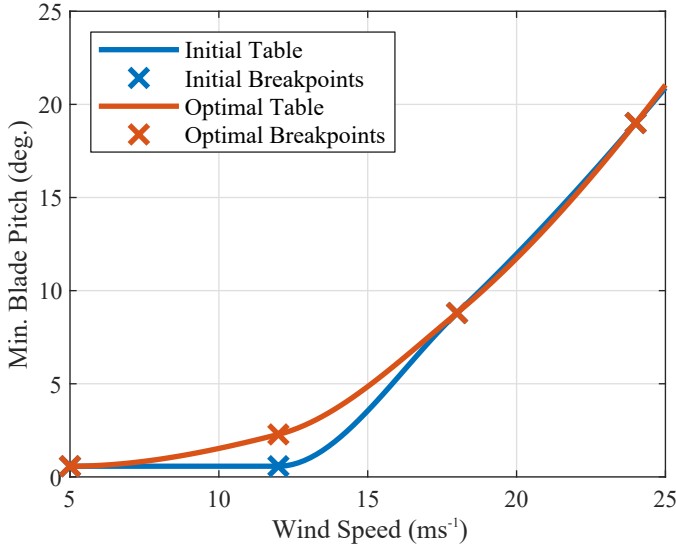

**Figure 8.** Minimum pitch setting as a function of wind speed. Each active breakpoint is tuned in a series of one-dimensional parameter optimizations. There is an additional high wind speed breakpoint at $50\ \text{ms}^{-1}$.





---

**Algorithm 1** Optimize Minimum Pitch Lookup Table

---

1: **Initialize:** Start with an initial guess for the pitch lookup table: $\theta_{\min}(u_i)$, e.g. in Table 4. Set starting maximum load $M_{u_i}(z^0) = -\infty$, for all actively optimized breakpoints $i = \{1, \ldots, N_{bp}\}$, where $N_{bp}$ is the number of wind speed breakpoints.

2: **for** Each breakpoint $u_i \in U$, $i = \{1, \ldots, N_{bp}\}$ **do**

3:      **Step 1:** Simulate $N_{seeds}$ random turbulent seeds with a mean wind speed of $u_i$ and the initial lookup table. Find the worst-case seed $n_{max}$ with the maximum load $M_{u_i}(z^0)$ over all seeds at breakpoint $u_i$; the starting power $\bar{P}_{u_i}(z^0)$ is the mean generator power of this simulation.

4:      **if** $M_{u_i}(z^0) < M_{u_j}(z^0)$, $j = \{1, \ldots, N_{bp}\} \setminus i$ **then**

5:          Skip **Step 2** and **Step 3**

6:      **end if**

7:      **Step 2:** Initialize and perform zeroth-order parameter optimization using the parameters in Table 4 and the cost function in (34), where

8:      $M(z^0) \leftarrow M_{u_i}(z^0)$

9:      $\bar{P}(z^0) \leftarrow \bar{P}_{u_i}(z^0)$

10:      Hard bounds on the pitch setting at the current breakpoint are set so that the minimum pitch table is non-decreasing: $\theta_{\min,LB} = \theta_{\min}(u_{i-1})$ and $\theta_{\min,UB} = \theta_{\min}(u_{i+1})$.

11:      The initial condition to the optimization procedure $z^1$ is set to enable an adequate search of the parameter space, i.e.

12:      $z^0 = \theta_{\min}(u_i)$

13:      **if** $z^0 < \theta_{\min,LB} + \mu$ **then**

14:          $z^1 = \theta_{\min,LB} + \mu$

15:      **else if** $z^0 > \theta_{\min,UB} - \mu$ **then**

16:          $z^1 = \theta_{\min,UB} - \mu$

17:      **else**

18:          $z^1 = z^0$

19:      **end if**

20:      The zeroth-order optimization procedure described in Section 2 is used to find

$$\theta_{\min}(u_i) = \arg\min_z C(z), \tag{35}$$

     where $z = \theta_{\min}(u_i)$ is the optimization parameter and the cost function $C(z)$ is defined in (34).

21:      **Step 3:** (Optional) Re-check the random turbulent seeds $n = \{1, \ldots, N_{seeds}\}$ using the new optimal pitch table and compute maximum load of each $M_r[n]$, finding the new worst-case seed $n_r$.

22:      **if** There is a new worst case: $n_r \neq n_{max}$ **then**

23:          Return to **Step 2** with $z^0 = \theta_{\min}(u_i)$ from (35).

24:      **end if**

25: **end for**

---





The algorithm (presented in Algorithm 1) is initialized by choosing an initial lookup table for the minimum blade pitch, $\theta_{\min}(u_i)$ in Table 4. In Step 1, $N_{\text{seeds}}$ random seeds are simulated to find the worst-case seed $n_{\max}$ with the maximum load at that breakpoint $M_{u_i}(z^0)$. The optimization procedure in Step 2 is only performed if the current breakpoint has a problematic peak load: one that is greater than loads seen at the other mean wind speeds (Line 4 of Algorithm 1). The starting loads are initialized to $M_{u_i}(z^0) = -\infty$ so that at least the first active break point is optimized. At the low wind speed breakpoint $u_0 = 5 \text{ ms}^{-1}$, the minimum pitch angle is set to the aerodynamically optimal angle $\theta_{\text{fine}}$; and at the high wind speed breakpoint $u_{N_{\text{bp}}+1} = 50 \text{ ms}^{-1}$, the optimal minimum pitch angle is set to the feather pitch angle. Neither $u_0$ nor $u_{N_{\text{bp}}+1}$ is an actively optimized breakpoint; they are, however, used as lower and upper bounds for the first $u_1$ and last $u_{N_{\text{bp}}}$ active breakpoints.

In Step 2, the initial guess that is used by the optimization procedure ($z^1$) is offset from the lower and upper bounds by the sample search area $\mu$ (lines 11 – 19 of Algorithm 1). Step 3 is optionally performed to re-check the other random turbulent seeds using the new, optimized minimum pitch lookup table. In some cases, a different random seed will have a peak load that exceeds that of the wind input that was originally optimized; if this is the case, Step 2 is repeated up to three times, using the previously optimized pitch angle as a lower bound.

**Table 4.** Parameters used to optimize the minimum pitch lookup table.

|  | Parameter | Variable | Value |
|---|---|---|---|
| Cost model | Load importance | $\kappa$ | 0.01 |
| Load simulations | Number of turbulent seeds at each wind speed | $N_{\text{seeds}}$ | 6 |
| Stage & | Number of stages | $N_{\text{stage}}$ | 4 |
| sample size | Samples per stage | $J$ | 2 |
| | Base step size | $\alpha_0$ | 1500 |
| Step | Armijo decrement factor | $\beta$ | 0.5 |
| size | Armijo threshold | $\sigma$ | 0.05 |
| | Max. size iterations | $k_{\max}$ | 3 |
| Newton's method approximation | | $D$ | 1 |
| Sample search radius & smoothing parameter | | $\mu$ | 0.5 |
| Initial | Number of actively optimized breakpoints | $N_{\text{bp}}$ | 3 |
| lookup | Total lookup-table breakpoints | $U$ | $\{5, 12, 18, 24, 50\}$ ms$^{-1}$ |
| table | Min. pitch setting at breakpoints | $\theta_{\min}(u_i)$ | $\{\theta_{\text{fine}}, \theta_{\text{fine}}, 9, 19, 90\}$ deg. |

Algorithm 1 is used to optimize the minimum pitch table in Fig. 8 using the parameters in Table 4 and the SUMR-13A wind turbine model. Six random turbulent seeds are initially simulated and only the 12 ms$^{-1}$ breakpoint requires optimization (the peak loads of the 18 and 24 ms$^{-1}$ simulations are all less than the 12 ms$^{-1}$ loads). $N_{\text{stage}} = 4$ stages with $J = 2$ samples-per-stage are used to optimize the 12 ms$^{-1}$ breakpoint; the results of the procedure are depicted in Fig. 7. The increased



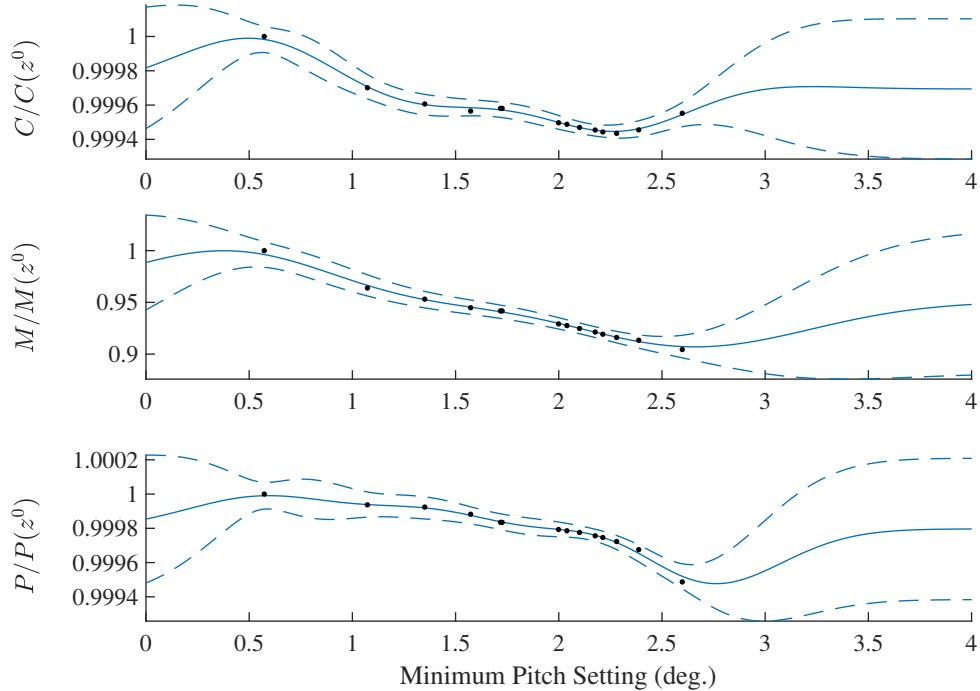

**Figure 9.** Uncertainty, via (19), associated with minimum pitch setting optimization: cost relative to initial guess $C/C(z^0)$, load relative to initial guess $M/M(z^0)$, and power relative to initial guess $P/P(z^0)$.

minimum pitch setting results in a 10.9% decrease in the tower load, a 8.51% decrease in the maximum blade load, and only a 0.03% decrease in the average generator power. These results hold for the full set of DLC 1.2 simulations (International Electrotechnical Commission, 2005).

# 4 Generalized Design Procedure

5   In this section, we present guidelines for performing similar optimization procedures. Experience gained in problem formulation, the usefulness of performing a preliminary "offline" analysis, and determining the parameters of the solver is shared.

## 4.1 Determine Problem and Goals

Using the zeroth-order optimization procedure described in this article for determining control parameters through simulation requires effort in setting up the problem and developing software. In order to justify the up-front effort, the task would ideally
10   be one that is repeated for many different rotor models, like the examples in Section 3. A task that is repeatedly performed also allows the designer to gain a deeper understanding for how control inputs (gains, parameters) affect simulation outputs of the wind turbine.





It is important to determine how the turbine should be simulated in order to generate the measures that are used for the optimization; they should highlight some problematic or indicative case that the control solution is trying to solve. For example, when optimizing the torque control gain for below-rated operation in Section 3.1, a below-rated wind field should be used and the power should be used as the cost function. The cost function should reflect the goals of wind turbine design (e.g. increasing

power capture or decreasing loads), have a basis in the reality of wind turbine operation (e.g. using gains that provide a stable control input), and also have a feasible solution. The optimization procedure presented in this article is only useful if the cost function represents the design goal, is represented well by the simulation information, and is simulated in a realistic environment.

## 4.2  Perform Preliminary Analysis

It is often helpful to perform a preliminary "offline" analysis to fine tune the cost function and optimization parameters. In an offline analysis, a grid search of the optimization parameter is used to estimate the output space of the simulations (e.g., maximum generator speed and blade loads), using a linear or quadratic estimate of the cost function. To clarify, the results in Section 3 are of "online" optimizations, where actual simulation data is used to compute the cost function and perform the optimization procedure. While one of the goals in developing this optimization procedure is to eliminate the large number of

simulations associated with grid searching, a grid search does help fine-tune the parameters of future, similar control tuning procedures that use zeroth-order optimization. If multiple measures are used in the cost function (e.g., in the pitch control tuning of Section 3.2), it is important to determine whether the cost function has a minimum within the parameter bounds. Otherwise, the cost function must be further refined. A preliminary offline analysis can be used to more quickly determine the optimization parameters (e.g., step size or smoothing parameter) that converge in the fewest number of simulations to some

"ground truth" determined from the estimated cost function determined using the initial grid search.

## 4.3  Set Simulation Parameters

As the examples of Section 3 illustrate, each optimization procedure requires slightly different parameters. While the parameters presented in Tables 1, 2, and 4 may not necessarily be the best ones, they have been fine-tuned through extensive offline testing and evaluating "online" tests that use actual simulation data as the measures used in the cost function. The goal of this

section is to provide general guidelines and rules-of-thumb, where possible, for choosing the parameters of the optimization procedure.

### 4.3.1  Sample search range and Newton's approximation

The smoothing parameter $\mu$ should be based on the optimization parameter $z$. The sample $z + \phi\mu$ should result in an adequate change to the cost function so that "good" gradients can be used for the descent algorithm; note that the magnitude of the

direction $||\phi|| = 1$. From the examples in Section 3, a different $\mu$ is required because the cost function of each application has a different magnitude and changes at different rates. Too large of a $\mu$ can result in samples that violate the hard bounds or



gradients that do not represent the local gradient at the stage sample. On the other hand, a $\mu$ that is too small can result in noisy gradients, the result of possibly non-smooth simulation information for samples that are close to each other.

When optimizing over multiple parameters, the $D$ matrix is used to approximate Newton's method for optimization. $D$ increases or decreases the descent direction $d_r$ where the sensitivity of the cost function to that parameter (dimension) is
small or large, respectively. Ideally, the matrix $D$ incorporates second-order information to scale the gradient estimate in each dimension. In a true Newton's method, where second-order information is available, $D = [\mathbf{H}\mathcal{C}(z)]^{-1}$, where $\mathbf{H}\mathcal{C}(z)$ is the Hessian of the cost function $\mathcal{C}$ at the parameter $z$. To approximate Newton's method, we use $D = \mathrm{diag}([D_1, \ldots, D_M])$, where

$$D_i \approx \left( \frac{\partial^2 \mathcal{C}}{\partial z(i)^2} \right)^{-1} \tag{36}$$

and $z(i)$ is the $i$th element of the parameter set $z$. The elements $D_i$ of $D$ can be determined from offline simulation analysis,
where (36) can be estimated by finding a quadratic regression of the cost space. Alternatively, $D_i$ can be manually tuned, i.e., if the dimension $i$ is not being adequately searched, $D_i$ should be increased.

For example, in the pitch controller tuning (Section 3.2), the cost function (shown in Fig. 3) is less sensitive to the damping ratio $z(2) = \zeta_{\mathrm{reg}}$ than it is to the natural frequency $z(1) = \omega_{\mathrm{reg}}$, so we use $D = \mathrm{diag}([0.25, 1])$. If only the first-order (estimated) information were used and the direction of the maximum gradient were exactly followed, the solution would zigzag in the $\omega_{\mathrm{reg}}$
direction and take longer to converge to the optimal solution in both the $\omega_{\mathrm{reg}}$ and $\zeta_{\mathrm{reg}}$ directions.

### 4.3.2 Step Size

The initial step size $\alpha_0$ is an important parameter to test offline and also fine tune when using online simulations to compute the gradient. It was found that for all optimization examples in this article, the product of the initial step size and the norm of the gradient should be on the order of a magnitude of 1, namely

$$\left\| \frac{\partial \mathcal{C}}{\partial z} \right\| \cdot \alpha_0 \approx 0.5 \text{ to } 1.5. \tag{37}$$

The parameters used in the Armijo step size rule were the same for all examples. Conservative values were used, which essentially only ensures a non-increasing cost function without a requirement on the rate of descent of the cost function.

### 4.3.3 Stages and Samples-per-Stage

Enough stages should be evaluated so that the cost function converges to some value; this is typically learned through offline
analysis or by trial-and-error in online tests. For example, when analyzing the pitch control tuning results of Fig. 4, the results suggest that the procedure could be performed with fewer stages, whereas it seems more stages could be used in the minimum pitch control tuning of Fig. 7. In general, it is found that fewer samples-per-stage (along with more stages) result in the fastest convergence with respect to the total number of simulations.





### 4.3.4 Parameter Bounds and Initial Guess

Hard constraints on the parameter should reflect the set of feasible parameters for the control task being optimized. However, the bounds should not be so small as to restrict the space and possibly miss non-obvious control solutions. The initial guess provided to the algorithm should also allow for the space to be adequately searched.

## 4.4 Perform Optimization and Evaluate Visualization

After performing initial, offline analysis and running the zeroth-order optimization algorithm using online simulation data, the whole procedure should be evaluated with the following questions:

1. Does the algorithm converge to a feasible solution?

2. Does the optimized parameter appear to be near the minimum of the visualized cost over the parameter space?

An affirmative answer to both of these questions should provide confidence in the optimized result.

## 5 Conclusions

In this article, we developed a data-driven approach for optimizing controller parameters using simulation results. By using a zeroth-order optimization algorithm, random samples are generated near an initial guess, which are used to compute the local gradient. A standard gradient descent method ensues, where a step size rule is used to ensure convergence and attempt to decrease in the cost function before the next guess is chosen and the process is repeated. We also use ordinary kriging to visualize the design space and its uncertainty to provide a level of confidence in the optimized result.

The zeroth-order algorithm was applied to three different applications in wind turbine control. To demonstrate the process on a one-dimensional parameter optimization, the torque control gain was tuned to optimize power capture in below-rated operation. The baseline pitch controller parameters were tuned in a two-dimensional optimization problem with the goal of minimizing structural loads and includes a constraint on the maximum generator speed. Using an adaptable cost function and step size, the algorithm was able to tune the baseline rotor speed control for rotors ranging from 40 to 400 meters in diameter. We compare the results, in terms of accuracy, convergence, and number of function evaluations (simulations) for different optimization parameters and against the standard grid search method. In a series of one-dimensional parameter optimizations, we also determined the settings of a lookup table for the minimum pitch limit of the pitch controller, reflecting the overall blade design process and system-level goals.

Since each optimization procedure depends on the specific control problem, we have provided a set of guidelines based on the experience gained during this study for developing future, similar optimization procedures. The methods presented in this article automate a usually manual process, reduce designer effort, and require fewer simulations compared with grid searching methods. These methods can be used for repeatable control tuning processes that are required for continually updating designs that must be evaluated in simulation using a well-functioning controller.



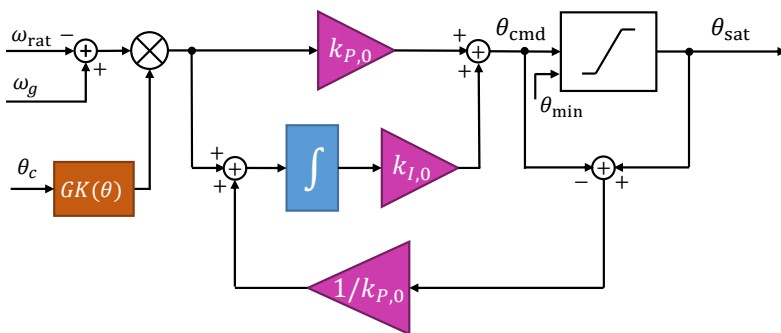

**Figure A1.** Proportional-integral control with anti-windup scheme used for above-rated control. The difference between the pitch control setpoint $\omega_\text{rat}$ and the generator speed $\omega_g$ is multiplied with the gain-correction factor $GK(\theta_\text{avg})$, which is a function of current collective blade pitch $\theta_c$. The proportional-integral gains at zero pitch, $k_{P,0}$ and $k_{I,0}$, are derived in (A15) and (A16). The pitch command $\theta_\text{cmd}$ is saturated to some minimum pitch setting $\theta_\text{min}$ and the output $\theta_\text{sat}$ is the input to the blade pitch actuator.

## Appendix A: Generalized Baseline Rotor Speed Pitch Controller

The pitch controller described in Sections 3.2 and 3.3 is based on the controller presented in the NREL-5MW reference manual (Jonkman et al., 2009); this standard control scheme is widely used as a reference for comparing control schemes and evaluating different aspects of turbine design. As shown in Fig. A1, the controller is a gain-scheduled proportional-integral (PI)
controller. The PI control architecture allows the generator speed dynamics to be represented as a $2^\text{nd}$-order system. Since the sensitivity of aerodynamic torque to blade pitch changes with the blade pitch, the PI gains are scheduled on the blade pitch. The pitch command is saturated to some minimum setting to control power or reduce blade loads; thus, an anti-windup scheme is necessary.

### A1   Regulator Mode and PI Gains

To derive the PI gains for a generic rotor model, a rigid model of the drivetrain is used:

$$\dot{\omega}_g = \frac{G}{J_\text{tot}}(\tau_a - G\tau_g), \tag{A1}$$

where $\omega_g$ is the generator speed, $J_\text{tot}$ is the total drivetrain inertia, including the rotor and generator components, $G$ is the gearbox ratio between the low-speed rotor shaft and the high-speed generator shaft, $\tau_a$ is the aerodynamic rotor torque caused by the wind and controlled via blade pitch, and $\tau_g$ is the generator torque, which is a control input. The rotor torque is non-
linearly dependent on the blade pitch $\theta$. The linearization with respect to a perturbation in blade pitch $\delta\theta$ is

$$\delta\dot{\omega}_g = \frac{G}{J_\text{tot}} \frac{\partial \tau_a}{\partial \theta} \delta\theta, \tag{A2}$$



where the differential torque $\delta\tau_g = 0$ because the torque is constant in above-rated operation. The sensitivity of the aerodynamic torque to rotor speed ($\frac{\partial\tau_a}{\partial\omega}$) is omitted since it has a much smaller magnitude than $\frac{\partial\tau_a}{\partial\theta}$. In terms of power $P$:

$$\tau_a = \frac{P(\theta)G}{\omega_g} \Rightarrow \left.\frac{\partial\tau_a}{\partial\theta}\right|_{\omega_g=\omega_{\text{rat}}} = \frac{G}{\omega_{\text{rat}}}\frac{\partial P}{\partial\theta}, \tag{A3}$$

where $\omega_{\text{rat}}$ is the rated generator speed, which is a constant operating point since it is the desired set-point of the controller. The

proportional-integral control is

$$\delta\theta = k_P\delta\omega_g + k_I\int\delta\omega_g dt, \tag{A4}$$

where $k_P$ and $k_I$ are the proportional and integral control gains, respectively, and $\delta\omega_g$ represents a generator speed perturbation.

By defining a new state, $\dot{\phi} = \delta\omega_g$, and combining equations (A2), (A3), and (A4), the generator speed dynamics are

$$J_{\text{tot}}\ddot{\phi} + \frac{1}{\omega_{\text{rat}}}\left(-\frac{\partial P}{\partial\theta}\right)k_P G^2\dot{\phi} + \frac{1}{\omega_{\text{rat}}}\left(-\frac{\partial P}{\partial\theta}\right)k_I G^2\phi = 0, \tag{A5}$$

which can be represented by a second-order dynamic system in the form of

$$M_{\text{reg}}\ddot{\phi} + D_{\text{reg}}\dot{\phi} + K_{\text{reg}}\phi = 0, \tag{A6}$$

where $M_{\text{reg}}$, $D_{\text{reg}}$, and $K_{\text{reg}}$ are the mass, damping, and stiffness of the "regulator mode," respectively. Alternatively, the regulator mode can be represented by its natural frequency $\omega_{\text{reg}}$ and damping ratio $\zeta_{\text{reg}}$, defined by

$$\omega_{\text{reg}} = \sqrt{\frac{K_{\text{reg}}}{M_{\text{reg}}}} \quad\text{and}\quad \zeta_{\text{reg}} = \frac{D_{\text{reg}}}{2\omega_{\text{reg}}M_{\text{reg}}}. \tag{A7}$$

By defining the desired properties of the generator speed dynamics, $\omega_{\text{reg}}$ and $\zeta_{\text{reg}}$, the proportional and integral gains are defined as follows:

$$k_P = \frac{2J_{\text{tot}}\omega_{\text{rat}}\omega_{\text{reg}}\zeta_{\text{reg}}}{G^2\left(-\frac{\partial P}{\partial\theta}\right)} \tag{A8}$$

and

$$k_I = \frac{J_{\text{tot}}\omega_{\text{rat}}\omega_{\text{reg}}^2}{G^2\left(-\frac{\partial P}{\partial\theta}\right)}. \tag{A9}$$

### A1.1   Power-Pitch Sensitivity and Gain Scheduling

Both the proportional (A8) and integral (A9) gains depend on the sensitivity of power to blade pitch

$$\left.\frac{\partial P}{\partial\theta}\right|_\theta = S(\theta), \tag{A10}$$

which we will define as $S(\theta)$ because it is a function of the blade pitch. Simulations in FAST are used to determine the pitch operating points at various above-rated wind speeds. The operating points are used in FAST linearizations, with all





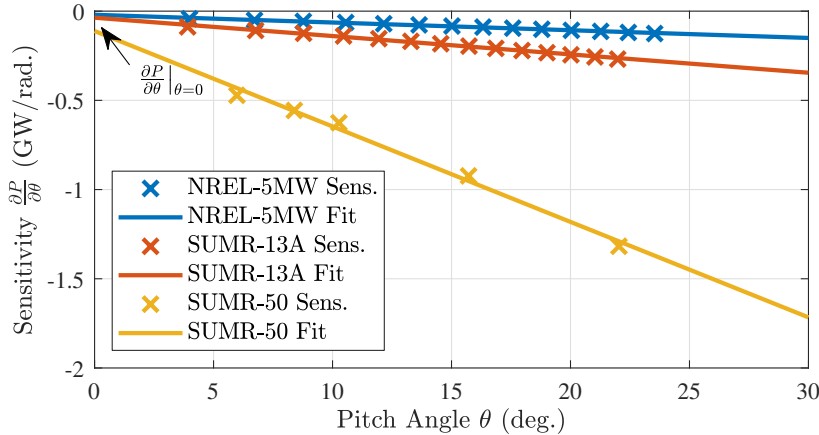

**Figure A2.** Sensitivity of power to pitch used for gain scheduling the pitch controller for a selection of rotors in this article. The sensitivity values obtained from FAST linearizations are fit linearly and used to determine the gain-scheduling parameters for the pitch controller.

the degrees-of-freedom disabled, providing the input-output sensitivity from pitch to power. The results of performing this sensitivity analysis for several rotors is shown in Fig. A2.

Because of the near-linear relationship with blade pitch $\theta$, the sensitivity can be parameterized by

$$S(\theta) = \left[\frac{S(0)}{\theta_k}\right]\theta + S(0) = S(0)\left(1 + \frac{\theta}{\theta_k}\right), \tag{A11}$$

where $S(0)$ is the sensitivity at $\theta = 0$ degrees and $\theta_k$ is the pitch angle at which the sensitivity doubles:

$$S(\theta_k) = 2S(0). \tag{A12}$$

From simulation results like those in Fig. A2, the parameters in (A11) can be estimated; they are used to define the gain-correction factor

$$GK(\theta) = \frac{1}{1 + \frac{\theta}{\theta_k}} \tag{A13}$$

and the final, gain-scheduled PI gains:

$$k_P = k_{P,0}GK(\theta) \quad \text{and} \quad k_I = k_{I,0}GK(\theta), \tag{A14}$$

where

$$k_{P,0} = \frac{2J_{\text{tot}}\omega_0\omega_{\text{reg}}\zeta_{\text{reg}}}{G^2\left[-S(0)\right]} \tag{A15}$$

and

$$k_{I,0} = \frac{J_{\text{tot}}\omega_0\omega_{\text{reg}}^2}{G^2\left[-S(0)\right]}. \tag{A16}$$

Fig. A1 depicts the implementation in block diagram form.



### A1.2   Summary of Pitch Control Tuning Procedure

To derive the parameters from simulations and tune the regulator mode, we use the following procedure:

1. Simulate the operating points in FAST using a steady wind input across above-rated wind speeds. Choose a large enough $S(0)$ so that the PI gains produce a stable result. Simulate for enough time for the values to reach steady state and record the blade pitch at each wind speed.

2. Linearize the turbine in FAST, disabling all of the degrees of freedom, at the wind speeds and pitch angles found from the previous step. Use the element of the input-output ($D$) matrix that corresponds to the pitch input and power output matrix to determine the sensitivity of power to pitch at the various pitch angle operating points. Plot the values and fit the parameters $S(0)$ and $\theta_k$ as in Fig. A2.

3. Tune the regulator mode ($\omega_{\text{reg}}, \zeta_{\text{reg}}$) using the desired design measure. Usually, larger natural frequencies ($\omega_{\text{reg}}$) result in better generator regulation, but also higher structural loads. A grid search could be used or an optimization procedure like the one in Section 3.2.

### Appendix B:  Turbine model summary

The turbine models summarized in Table B1 were used to perform the control tuning optimization procedures detailed in this article.

*Competing interests.*   The authors declare no competing interests.

*Acknowledgements.*   The information, data, or work presented herein was funded in part by the Advanced Research Projects Agency - Energy (ARPA-E), U.S. Department of Energy, under Award Number DE-AR0000667. Support from a Palmer Endowed Chair Professorship is also gratefully acknowledged. The views and opinions of the authors expressed herein do not necessarily state or reflect those of the United States Government or any agency thereof.





**Table B1.** Summary of turbine models used in this study.

| Turbine Model | SUMR-D | NREL-5MW | CONR-13 | SUMR-13A |
|---|---|---|---|---|
| Rated Power | 54.5 kW | 5 MW | 13.2 MW | 13.2 MW |
| Rated Rotor Speed | 21.5 rpm | 12.1 rpm | 7.44 rpm | 9.90 rpm |
| Rated Wind Speed | 5.05 ms$^{-1}$ | 11.3 ms$^{-1}$ | 11.3 ms$^{-1}$ | 11.3 ms$^{-1}$ |
| Hub Height | 34.86 m | 87.0 m | 142.4 m | 142.4 m |
| Rotor Radius | 22.8 m | 63.0 m | 102.5 m | 101.2 m |
| Rotor Position | Downwind | Upwind | Upwind | Downwind |
| Blade Mass | 997 kg | 17.7 Mg | 49.5 Mg | 51.8 Mg |
| Number of Blades | 2 | 3 | 3 | 2 |
| Max Chord | 1.56 m | 4.65 m | 5.23 m | 7.22 m |
| Cone Angle | 12.5 deg. | -2.5 deg. | -2.5 deg. | 12.5 deg. |

| Turbine Model | SUMR-13B | SUMR-13C | SUMR-25 | SUMR-50 |
|---|---|---|---|---|
| Rated Power | 13.2 MW | 13.2 MW | 25 MW | 50 MW |
| Rated Rotor Speed | 7.99 rpm | 6.87 rpm | 6.13 rpm | 4.19 rpm |
| Rated Wind Speed | 10.3 ms$^{-1}$ | 9.30 ms$^{-1}$ | 10.5 ms$^{-1}$ | 10.3 ms$^{-1}$ |
| Hub Height | 142.4 m | 168 m | 210 m | 280 m |
| Rotor Radius | 125.4 m | 145.9 m | 171.9 m | 239.7 m |
| Rotor Position | Downwind | Downwind | Downwind | Downwind |
| Blade Mass | 83.2 Mg | 105 Mg | 127 Mg | 426 Mg |
| Number of Blades | 2 | 2 | 2 | 2 |
| Max Chord | 6.79 m | 9.29 m | 10.7 m | 16.6 m |
| Cone Angle | 12.5 deg. | 12.5 deg. | 12.5 deg. | 12.5 deg. |

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
