# Peer review of "Automatic controller tuning using a zeroth-order optimization algorithm"

_Wind Energy Science, 2020_

## Referee Comment (RC1) · Alan Wai Hou Lio (Referee) · 22 Apr 2020

The authors present an automated procedure for controller tuning for wind turbines. The procedure is based on the zeroth-order optimisation algorithm and many turbine simulations, subsequently, the optimal set of control parameters is computed. The procedure is applied to three different wind turbine control problems - generator torque control, blade pitch control and minimum pitch settings.

The paper is well-written. Automatic tuning of control parameters is definitely an interesting inter-disciplinary area, requiring expertise in optimisation, control and wind energy. I believe this work is relevant to the community, as the turbine size is getting better nowadays and the flexible turbine dynamics might not be fully captured in a

simplified model for model-based design. I listed some comments below.

Comments:

p1 l20: 'The wind turbine control tuning procedure is normally a manual process and often requires expert knowledge of the controller and turbine operation.'. This statement is not necessarily true. As the authors later pointed out in Appendix A, most of the tuning process is automatic and based on a simplified second-order rotor drive-train model. Can the authors elaborate on the term 'manual process'?

p1 l24: 'The simplest method to tune a controller using simulation information is to exhaustively search the design space and then make an educated design choice of the parameter.' I agree this might be a 'layman' approach but I think most of the people would use a model-based approach as described in your reference (Johnson et al. 2009). Your introduction section is written like for controller tuning that always requires solving an optimisation problem. Can the authors introduce the commonly used model-based methods (e.g. Johnson et al. 2009) in your introduction section?

p2 l26: 'Our approach to sampling the parameter space is based on stochastic approximation or "zeroth-order optimization,"...'. Can the authors justify why you choose this zeroth-order optimisation technique rather other optimisation techniques?

p3 l5: 'Instead of using a single cost function that attempts to account for all aspects of wind turbine design...'. Is this part of the sentence necessary? Can the authors provide the references that use a single cost function for the whole design?

p6 l9 & p7 l27: It seems the N_stage is used to determine the termination of the iterations. How did the authors determine this N_stage?

p9 Figure1: How many simulations did the authors run to generate the solid blue line?

p9 l5: 'The algorithm finds the optimal kfact and converges in 7 stages'. Is the number of stages determined by the settling function in (21)?

p9 l7: 'average power is increased by 0.67%, compared to kfact = 1': The power is indeed increased. Did the authors check other loads such as tower, blade and shaft torsion?

p10 l14: What is M(z)? structural loading of what components?

p12 table 2: Why N_stage are 7 and 12?

p14 table3: The M(z) are tower and max blade loads. Did the authors check the rotor speed performance? Also the shaft torsion and pitch activities? Why are these variables not considered in this case?

p15 Figure 4: How many simulations did the authors run to get the colour contour in the left of Figure 4?

p17 l14: 'However, we believe that a controller that constrains extreme events and maximizes power capture better reflects the overall wind turbine design goals.' Can the authors elaborate a bit on this statement? Does the statement imply that in the above-rated region, the regulation of rotor speed is less important as long as the rotor speed does not exceed its max value? Did the authors also investigate on loads of other turbine components to support this statement?

p18 l15: '...quantifying the relative importance between reducing peak loads...'. Which peak loads were the authors indicating?

p18 Section 3.3. This section investigates the optimal minimum pitch values in terms of loads and power. Would this optimal minimum pitch angle affect the result in Section 3.1 and 3.2? How did the authors ensure these tunings were decoupled? Wouldn't it be better if one single multi-objective optimisation problem is solved instead of three?

p20 Figure 7: What turbine is this? Is this not the NREL 5MW? The power seems much bigger than 5MW.

---

## Referee Comment (RC2) · Luca Sartori (Referee) · 28 Apr 2020

This paper deals with a novel procedure for the automatic tuning of a wind turbine control system. In the presented approach, results from dynamic simulations are used to feed an optimization method which automatically designs specific control parameters. The whole procedure relies on a zeroth-order algorithm which generates and resolve samples in the proximity of a certain guess, in order to identify the best descent direction.

In my opinion, this is a very well presented work. The topic of automatic controller optimization is certainly hot in wind energy, especially for very large turbines and I believe the procedure highlighted in this paper could help finding novel ways to improve

the performance of modern turbines.

The work is well organized and clear. The Authors give a detailed description of the algorithm as well as several examples in which the method successfully manages various control problems. In addition, results for several rotor sizes are given which contribute to extend the validity of their findings.

I have some comments here below:

Section 3.1:

-What simulations did you run at this step? Did you cover the whole range of speed below rated or just focused on a single wind speed? I raise this concern because, in highly flexible rotors, increasing wind speeds typically trigger increased torsional deformation, and I think your optimal solution could be somewhat influenced.

- I like your idea of testing the ZOO on separated control problems. This is very enlightening for what concerns the method, but I think it reduces the generality of the results. For example, in this application the optimization leads to an increase in power, but what about loads (and fatigue in particular)? In other words: if we consider a real application, would the power gain high enough to be convenient against (probably) higher loads?

- Perhaps I've missed something but, when you only require 2 samples for iteration aren't you basically going back to a standard gradient method with centered finite differences?

Section 3.2:

-This is, in my opinion, the weakest application (an interesting one, though!). it seems like the ZOO provides an expected solution (reducing the pitch frequency and increasing the damping), so that I'm not sure the results are actually worth the effort.

-In addition, a drastic reduction in the pitch actuation can have significant drawbacks: what about the torque? In my experience a slower pitch requires a faster torque to

avoid severe power losses. Then: what about the power?

-Another concern is this: how the 'slower' pitch reacts to dangerous situations (extreme gusts, faults and other conditions when a fast response is required)?

Section 3.2.2:

-I like the comparison between the ZOO and the grid search, however, i don't think your example gives the full picture. First, the solution of the grid search depends on the grid itself, and thus, I think in your example the only example which actually makes sense is the example with 10x10 samples. In this view, Figure 6(f) clearly shows that the 6x6 and 8x8 cases are not 'converged' to the real optimal solution. Looking at Figure 6(e) I dont' even think the 10x10 case led to the 'real' optimum. probably a further refinement of the mesh is required.

- Additionally, the effectiveness of the grid search depends on the complexity of the underlying function, so the example you provided gives only a partial conclusion. I would also like to see how the methods compare when tested on (perhaps) 'analytical' functions of growing complexity.

Section 3.3:

- I think this is the most interesting example, as it deals with a very common problem of control tuning. However, i don't personally like merit functions which are built from a linear combination of requirements, like in this case. What I don't like is that, such a linear combination requires to define the weighting parameters of the various terms in the merit function. In this case, for example, you arbitrarily chose to set k=0.01, and this led to an optimal curve. But, if you change that value, you'll probably obtain a family of curves. So, once again, what is the 'real' optimum?

Figure 4:

- Please add (a,b,c,d) identifier in the various subplots.

Sorry for being so verbose, but I enjoyed this paper and I would like to raise some points to further discuss your approach.

Thank you for this contribution.

---

## Author Comment (AC1) · 5 Jun 2020

**Response to Referee #1**

**MS Number:** wes-2020-63
**Title:** Automatic controller tuning using a zeroth-order optimization algorithm
**Corresponding author:** Daniel Zalkind

We would like to thank Dr. Alan Wai Hou Lio for the time and effort put in reviewing our submission and for the detailed comments. All the comments have been addressed in the revised version of the manuscript. The comments and suggestions provided by Dr. Lio have enabled us to improve the quality of the manuscript.

The following table collects the referee's comments, the individual responses to specific comments and suggestions, and the authors' changes in the manuscript. In addition, a color-coded version of the manuscript is provided, in which all changes can be identified. We have used the red color to indicate text that has been removed from the submitted manuscript. The descriptions in blue represent the added or re-written parts, addressing the referee's comments.

| *Comments of Referee #1* | *Authors' Responses* |
|---|---|
| Page 1, Line 20: 'The wind turbine control tuning procedure is normally a manual process and often requires expert knowledge of the controller and turbine operation.'. This statement is not necessarily true. As the authors later pointed out in Appendix A, most of the tuning process is automatic and based on a simplified second-order rotor drive-train model. Can the authors elaborate on the term 'manual process'? | **Answer:** Most of the tuning procedure in Appendix A is automated, except for the final step. Designers still must choose the natural frequency and damping ratio, and a particular choice may not apply equally well to different rotor designs. In this paper, our focus is to automate and optimize the design choices that were previously manually determined or based on previous design choices. |
| | **Changes in manuscript:** We have re-worded these statements to clarify that the whole process is not manual. We are trying to automate design choices that require simulation results: |
| | The wind turbine control tuning  can be automated, but design choices for the various parameters often require expert knowledge of the controller and turbine operation. An automated procedure to determine these choices could reduce the design cycle time of a manufacturer's research and development process or… |
| Page 1, Line 24: 'The simplest method to tune a controller using simulation information is to exhaustively search the design | **Answer:** Thank you for this comment, which has given us the opportunity to clarify potential merits of the proposed methodology. It is true that one may perform an exhaustive search over a properly discretized domain of the search space. |

| | |
|---|---|
| space and then make an educated design choice of the parameter.' I agree this might be a 'layman' approach but I think most of the people would use a model-based approach as described in your reference (Johnson et al.2009). Your introduction section is written like for controller tuning that always requires solving an optimization problem. Can the authors introduce the commonly used model-based methods (e.g. Johnson et al. 2009) in your introduction section? | However, there are two main disadvantages: 1) the computational burden rapidly increases with the number of search points (following an exponential growth), thus rendering this process computationally infeasible for fine discretizations; 2) in the case of coarse grids, the tuning may no longer be optimal, as the optimal configurations may be far from a point in the grid (in the worst case, half way between two points of the search grid). The proposed approach is *computationally lighter,* and it is able to provably *identify optimal configurations*. It is also worth pointing out that the proposed approach naturally lends itself to an adaptive tuning strategy in the case of changing operating conditions (where it may not be possible to include an exhaustive search) and can effectively cope with model mismatches. |
| | **Changes in manuscript:** We have updated these statements to reflect our meaning more directly:

The simplest method to  determine these design choices using simulation results is to exhaustively search the design space and then make an educated design choice of the parameter. However, exhaustive search may become computationally intractable for fine discretizations of the search space; on the other hand, coarse discretizations may lead to sub-optimal design choices.

And we elaborate on model-based pitch and torque controllers:

Model-based pitch (Hansen et al., 2005) and torque (Johnson et al., 2006) controllers usually result in functioning controllers, but require rules-of-thumb to determine the closed-loop characteristics and can be inaccurate when there are uncertainties in the model. |
| Page 2, Line 26: 'Our approach to sampling the parameter space is based on stochastic approximation or "zeroth-order optimization,"...'. Can the authors justify why you choose  -order optimization technique  rather other optimization techniques? | **Answer:** Thank you for this comment. We selected a zeroth-order optimization to bypass the need for an analytical model to compute gradients (for classical first-order methods) or Hessian matrices (for second-order methods). In particular, the proposed method estimates first-order information from functional evaluations (e.g., simulations). This is useful in many applications when the available model is inaccurate, or a model is not available. |
| | **Changes in manuscript:** We elaborate on our choice:

Other optimization algorithms require an analytical model; the proposed method relies on functional evaluations (e.g., simulation data) and does not require a model to compute gradient information. |

| | |
|---|---|
| Page 3, Line 5: 'Instead of using a single cost function that attempts to account for all aspects of wind turbine design...'. Is this part of the sentence necessary? Can the authors provide the references that use a single cost function for the whole design? | **Answer:** Maybe not. We were trying to avoid solving problems related to the overall turbine performance, which are often solved with aerodynamic or structural design, but instead solve more specific control problems. |
| | **Changes in manuscript:** We have changed our wording to reflect our meaning more directly:

Instead of using  cost functions directly related to overall wind turbine performance, our work solves specific wind turbine control problems that are  related to the cost of energy. |
| Page 6, Line 9 and Page 7, Line 27: It seems the N_stage is used to determine the termination of the iterations. How did the authors determine this N_stage? | **Answer:** There are different ways to determine N_stage. For example, one way is to check the norm of the estimated gradient at the current point; another way can be dictated by a given budget on the total number of simulations that are to be performed. We analyze the performance versus the number of stages N_stages in Section 3.2.2. |
| | **Changes in manuscript:** We elaborate on how N_stage is a pre-selected parameter and how we have chosen it as well as discuss the issue with using a stopping condition for a zeroth-order optimization:

…where the number of stages $N_{stage}$ is determined before running the algorithm. A typical stopping condition involves checking whether the norm of the gradient is less than a given threshold or dictating a budget on the number of simulations that are to be performed. We investigate the performance versus $N_{stage}$ in Section 3.2.2. |
| Page 9, Fig. 1: How many simulations did the authors run to generate the solid blue line? | **Answer:** The cost estimate in each figure of the initial draft was determined after all simulations have been performed, but it can be shown earlier. While it would be interesting to show the evolution of the estimate, we unfortunately do not have space for it. |
| | **Changes in manuscript:** In Figures 1 and 3, when we show the first iteration of the zeroth-order optimization and the sampled cost function, we show the cost estimate determined after the first stage. In the caption of Figs. 1, 2, 3, 4, and 7, we clarify how many stages and simulations were used to determine the cost estimate. |

| | |
|---|---|
| Page 9, Line 5: 'The algorithm finds the optimal kfact and converges in 7 stages'. Is the number of stages determined by the settling function in (21)? | **Answer:** The settling function is used to analyze the cost at each stage, and it requires the final cost to compute, so it cannot be used as a stopping condition.

A typical stopping condition would be when there is a gradient with a magnitude less than some threshold, but the gradient information we have is estimated and imperfect, which could lead to the algorithm stopping prematurely. |
| | **Changes in manuscript:** We have re-worded this sentence so that it doesn't imply that the algorithm "finds" a solution in $x$ number of stages. We determine how many stages are adequate based on the results, which show that there are diminishing returns in terms of the cost reduction after 5 stages:

The algorithm finds close to the optimal k_fact  in 5 stages and realizes diminishing returns thereafter (Fig. 2b). |
| Page 9, Line 7: "…average power is increased by 0.67%, compared to k_fact = 1." The power is indeed increased. Did the authors check other loads such as tower blade and shaft torsion? | **Answer:** We did not consider loads when optimizing the energy production. More information about the wind turbine system would be required to determine the relative benefit of increasing power versus changes in loads at those wind speeds. |
| | **Changes in manuscript:** The clause was removed when new results were found that incorporate more wind speeds and a section (Section 3.4) was added to discuss systems engineering considerations, which we describe in more detail below. |
| Page 10, Line 14: What is M(z)? structural loading of what components? | **Answer:** $M(z)$ is meant to be any load or measure, which represents either peak blade load or tower fatigue in Section 3.2. |
| | **Changes in manuscript:** We have updated the text to better clarify $M(z)$ here:

In general,  changing the bandwidth of the pitch controller  via $\omega_{reg}$  alters the structural loading of various components, which we denote generically with $M$ in the following. In this section, we use $M$ to denote tower fatigue or peak blade loading, though any load could be used that results in a feasible optimization problem; the control and hardware designers must determine what loads are important to the overall turbine design. |
| Page 12, Table 2: Why are N_stage | **Answer:** We tested both to perform the analysis in Section 3.2.2 |

| | |
|---|---|
| 7 and 12? | and Fig. 6. |
| | **Changes in manuscript:** We have included an additional footnote below Table 2:

*In Section 3.2.2, we compare the performance of using different numbers of stages Nstage and samples per stage J. |
| Page 14, Table 3: The M(z) are tower and max blade loads. Did the authors check the rotor speed performance? Also, the shaft torsion and pitch activities? Why are these variables not considered in this case? | **Answer:** We determined the rotor speed performance in terms of overspeed. Shaft torsion and pitch activity were not considered because they were not critical during the blade design of our research turbine. We do know that tower fatigue and pitch actuation are closely linked. Shaft torsion and pitch actuation are similar, but we were most concerned with peak loads. Different measures could be used for $M(z)$ if the parameters that we are randomly sampling result in a change to the cost function during the simulations tested. |
| | **Changes in manuscript:** We mentioned this with the definition of $M(z)$:

In general,  changing the bandwidth of the pitch controller  via $\omega_{reg}$  alters the structural loading of various components, which we denote generically with $M$ in the following. In this section, we use $M$ to denote tower fatigue or peak blade loading, though any load could be used that results in a feasible optimization problem; the control and hardware designers must determine what loads are important to the overall turbine design. |
| Page 15, Fig. 4: How many simulations did the authors run to get the color contour in the left of Figure 4? | **Answer:** In Fig. 4, the cost estimate uses all of the simulation results available. After 7 stages, a total of 84 different parameter sets and simulations were performed. |
| | **Changes in manuscript:** We have added this information to the caption of Fig. 4. |
| Page 17, Line 14: 'However, we believe that a controller that constrains extreme events and maximizes power capture better reflects the overall wind turbine design goals.' Can the authors | **Answer:** In other work [1], we describe how turbine shutdowns result in a net AEP reduction and that in many large rotor designs, AEP is the primary driver to the cost of energy. Typical rotor speed controllers are designed to regulate to a fixed rotor speed with a high enough bandwidth so that the maximum generator speed constraint is not violated. Instead of focusing on a quantity |

| | |
|---|---|
| elaborate a bit on this statement? Does the statement imply that in the above-rated region, the regulation of rotor speed is less important as long as the rotor speed does not exceed its max value? Did the authors also investigate on loads of other turbine components to support this statement? | that measures how well the generator speed is regulated, we focus on whether or not the maximum generator speed constraint is violated. Loads are generally reduced with pitch actuation, but a more in-depth investigation of the global loads is left for future work.

[1] D. S. Zalkind and L. Y. Pao, "Constrained Wind Turbine Power Control," *2019 American Control Conference (ACC)*, Philadelphia, PA, USA, 2019, pp. 3494-3499, doi: 10.23919/ACC.2019.8814860. |
| | **Changes in manuscript:** We have summarized the answer above and included it in the manuscript.

Typical pitch controllers are designed to regulate the generator speed to some fixed rotor speed with a high enough bandwidth so that the maximum generator speed constraint is not violated. Instead of focusing on a quantity that measures how well the generator speed is regulated, we focus on whether or not the maximum generator speed constraint is violated. An initial investigation (Zalkind and Pao, 2019) of the loads on the other turbine components show a reduction in blade and low-speed shaft fatigue and pitch actuation, but a more in-depth loads investigation is left for future work. |
| Page 18, Line 15: '...quantifying the relative importance between reducing peak loads...'. Which peak loads were the authors indicating? | **Answer:** Thank you for this comment. We meant the blade flapwise peak load. |
| | **Changes in manuscript:** We have tried to further clarify this in the text:

…, M is the  maximum blade flapwise load (over all blades),… |
| Page 18, Section 3.3: This section investigates the optimal minimum pitch values in terms of loads and power. Would this optimal minimum pitch angle affect the result in Section3.1 and 3.2? How did the authors ensure these tunings were decoupled? Wouldn't it be better if one single multi-objective optimization problem is solved instead of three? | **Answer:** Thank you for this question. There is a coupling between the minimum pitch $\theta_{min}$ and the torque and the pitch control, but we believe the change in optimal parameters would be small. The $k\omega^2$ law is active when $\theta_{min}$ is close to or equal to $\theta_{fine}$. $\theta_{min}$ will change the peak blade and fatigue tower loads, the optimal pitch control parameters are not changed by a noticeable amount.

A multi-objective optimization problem is an interesting idea for future work. Several issues that need to be overcome are:
 - Several simulations would need to be run for each parameter set, since the control parameters do not affect all DLC simulations. |

- Changing multiple parameters simultaneously could make finding meaningful gradients more difficult, resulting in a greater computational cost to finding optimal parameters.

The additional complexity of a multi-objective optimization would have to outweigh the computational cost.

The simpler approach that we used was to solve the problems sequentially and (optionally) multiple times. We achieved the best results by solving each problem separately and then re-tuning. In other words, optimizing the torque control, then the pitch control, and then the minimum pitch setting, followed by a re-optimization of the torque and pitch controllers.

**Changes in manuscript:** We have summarized the answers to your questions in a discussion in Section 3.4: Coupling Between Control Optimizations and Systems Engineering Considerations

Changing the minimum pitch setting of the controller can have an effect on the below-rated power production (optimized in Section 3.1) and peak and fatigue loads (optimized in Section 3.2). Though this coupling exists, in our experience, the effect on the optimized parameters is small.

In future work, a multi-objective optimization might be more suitable, where all the tuning procedures are simultaneously performed. A potential challenge would be determining what simulations should be used to efficiently optimize all of the control parameters. Currently, each control tuning procedure requires running different simulations. Additionally, the goal of this work was to automate design choices, rather than having to choose from a set of possible choices that would result from a multi-objective optimization.

However, with additional resources, our goal could shift from efficient optimization of smaller problems to larger optimizations of the overall turbine system. Within a system engineering framework, more information might determine which simulations and loads are sensitive to parameter changes. In this article, we focused on minimizing the peak blade loads of the SUMR rotors because those were the design driving load of those blades. Other loads could certainly be used, but all loads are not important to the overall design: some components are over-designed, and others drive design; this information depends on the

specific design but could be determined using detailed system engineering tools.

Our goal was to reduce the design cycle times for processes that already occur during control design. Rather than solving all problems at once, we propose solving them in sequence, in the order they are presented: first optimizing the torque and pitch controllers, and then tuning the minimum pitch setting for peak loads. Then with the new minimum pitch table, a designer could optionally re-optimize the torque and pitch gains; we have done this and witness little change. Solving smaller problems tends to be more efficient in terms of the number of simulations and more transparent in terms of how control parameters affect different performance measures during specific simulations. We discuss setting up similar optimization problems for future work in Section 4.

| Page 20, Fig. 7: What turbine is this? Is this not the NREL 5MW? The power seems much bigger than 5MW. | **Answer:** This result was for the SUMR-13 turbine. It has been updated to the NREL-5MW for consistency with the other applications and to link with the discussion about coupling in the optimization procedures. |
| | **Changes in manuscript:** We mention the turbine model in the caption of Fig. 7. |

[revised manuscript text omitted]

---

## Author Comment (AC2) · 5 Jun 2020

**MS Number:** wes-2020-63
**Title:** Automatic controller tuning using a zeroth-order optimization algorithm
**Corresponding author:** Daniel Zalkind

We would like to thank Dr. Luca Sartori for the time spent in reviewing our submission and for the constructive feedback. We have addressed all the comments in the revised version. The comments and suggestions provided by Dr. Sartori have enabled us to clarify important aspects of the proposed methodology and to improve the overall quality of the manuscript.

The following table collects the referee's comments, the authors' responses to each point, and the authors' changes in the manuscript. In addition, a color-coded version of the manuscript is provided, in which all changes can be easily identified. We have used the red color to indicate text that has been removed from the submitted manuscript. The descriptions in blue represent the added or re-written parts, addressing the referee's comments.

| *Comments of Referee #2* | *Authors' Responses* |
|---|---|
| Section 3.1:
-What simulations did you run at this step? Did you cover the whole range of speed below rated or just focused on a single wind speed? I raise this concern because, in highly flexible rotors, increasing wind speeds typically trigger increased torsional deformation, and I think your optimal solution could be somewhat influenced. | **Answer:** In the initial draft, we simulated a single 8 m/s turbulent wind input, but you make a good suggestion. It is possible to run a 6, 8, and 10 m/s turbulent wind input and weight them using a wind speed distribution. We have tried this and found interesting results. |
| | **Changes in manuscript:** Section 3.1 was updated with this addition:

where the cost function, (eq 24), is the negative of the weighted average mean generator power, (eq 25), which uses the average generator power of a simulation with mean wind speed $u$ and $p(u)$ is the Weibull wind speed distribution. The optimization parameter $z = k_{fact}$, the Weibull shape and scale parameters are 2.17 and 10.3, respectively, and we used $U = 6$, 8, and 10 m/s to span the below-rated wind speeds.

Figure 2 shows the new results, which we comment on:

If only a single simulation at 8 m/s is used and the controller is exclusively in Region II, we find a lower optimal $k_{fact}$ than is shown in Fig. 2a (Zalkind et al., 2020). By including other wind speeds and the |

transition region, as shown in Figs. 2(c, d, and e), the optimal $k_{fact}$ is nearly 1.

| | |
|---|---|
| - I like your idea of testing the ZOO on separated control problems. This is very enlightening for what concerns the method, but I think it reduces the generality of the results. For example, in this application the optimization leads to an increase in power, but what about loads (and fatigue in particular)? In other words: if we consider a real application, would the power gain high enough to be convenient against (probably) higher loads? | **Answer:** Thank you for this comment. Our objective was to decouple the control problems, but we recognize that a complete decoupling may not be feasible. A change in power may be reflected in an increase in loads; however, without connecting the control tuning to an overall turbine optimization, the overall effect on cost is not known. We believe that changing the torque control parameter will not have a large effect on the fatigue loads, which are greatest near rated wind speeds and above rated.

Your comment prompted us to think about the design of cost functions for the optimization problem that may account for both power and loads; this will be part of our future research efforts. |

**Changes in manuscript:** In a new Section 3.4, we discuss the coupling of the control optimizations and the applicability of the process in system engineering tools:

Changing the minimum pitch setting of the controller can have an effect on the below-rated power production (optimized in Section 3.1) and peak and fatigue loads (optimized in Section 3.2). Though this coupling exists, in our experience, the effect on the optimized parameters is small.

In future work, a multi-objective optimization might be more suitable, where all the tuning procedures are simultaneously performed; the zeroth-order method is suitable in this case. A potential challenge would be determining what simulations should be used to efficiently optimize all of the control parameters. Currently, each control tuning procedure requires running different simulations. Additionally, the goal of this work was to automate design choices, rather than having to choose from a set of possible choices that would result from a multi-objective optimization.

However, with additional resources, our goal could shift from efficient optimization of smaller problems to larger optimizations of the overall turbine system. Within a system engineering framework, more information might determine which simulations and loads are sensitive to parameter changes. In this article, we focused on minimizing the peak blade loads of the SUMR rotors because those were the design driving load of those blades. Other loads could certainly be used, but all loads

| | are not important to the overall design: some components are over-designed, and others drive design; this information depends on the specific design but could be determined using detailed system engineering tools.

Our goal was to reduce the design cycle times for processes that already occur during control design. Rather than solving all problems at once, we propose solving them in sequence, in the order they are presented: first optimizing the torque and pitch controllers, and then tuning the minimum pitch setting for peak loads. Then with the new minimum pitch table, a designer could optionally re-optimize the torque and pitch gains; we have done this and witness little change. Solving smaller problems tends to be more efficient in terms of the number of simulations and more transparent in terms of how control parameters affect different performance measures during specific simulations. We discuss setting up similar optimization problems for future work in Section 4. |
|---|---|
| - Perhaps I've missed something but, when you only require 2 samples for iteration aren't you basically going back to a standard gradient method with centered finite differences? | **Answer:** Yes, for a 1-dimensional problem with 2 samples, eq (3) boils down to a gradient method where the gradient estimate is obtained via finite differences. |
| | **Changes in manuscript:** We mention this in Section 3.1, when defining the sample generation:

Eq (26),
which simplifies (3) to a centered finite difference approximation of the gradient for this 1-dimensional application. |
| Section 3.2:

-This is, in my opinion, the weakest application (an interesting one, though!). it seems like the ZOO provides an expected solution (reducing the pitch frequency and increasing the damping), so that I'm not sure the results are actually worth the effort. | **Answer:** The strength of this work is not in finding non-obvious solutions; rather, we proposed an automated process that is computationally lighter than a brute-force grid search, and it is able to provably identify optimal configurations. |
| | **Changes in manuscript:** We have re-phrased some of the introduction to more clearly state our goal of automating design choices, like the rotor speed pitch control:

The wind turbine control tuning  can be automated, but design choices for the various parameters often require expert knowledge of the controller and turbine |

| | |
|---|---|
| | operation. An automated procedure to determine these choices could reduce the design cycle time of a manufacturer's research and development process or… |
| -In addition, a drastic reduction in the pitch actuation can have significant drawbacks: what about the torque? In my experience a slower pitch requires a faster torque to avoid severe power losses. Then: what about the power? | **Answer:** We use a constant torque control above rated, so cannot comment on extra torque actuation. |
| | **Changes in manuscript:** We note that we are using a constant torque in the appendix: As shown in Fig. A1, the controller is a gain-scheduled proportional-integral (PI) controller with constant torque above rated. |
| -Another concern is this: how the 'slower' pitch reacts to dangerous situations (extreme gusts, faults and other conditions when a fast response is required)? | **Answer:** We do not believe that we "slow down" the pitch response as much as increase the damping of the response and reduce the oscillations in the resulting pitch control. A pitch controller with this response has been used for many rotor designs, but with a focus on the power producing DLCs. During DLC 1.4, which is usually the most problematic, the initial response, due to the proportional gains, is similar between a standard controller and the one we present as optimal. During fault events, we assume supervisory control would override this controller. |
| | **Changes in manuscript:** We discuss the difference between our optimized control gain and the original at the end of Section 3.2.1: Typical pitch controllers are designed to regulate the generator speed to some fixed rotor speed with a high enough bandwidth so that the maximum generator speed constraint is not violated. Instead of focusing on a quantity that measures how well the generator speed is regulated, we focus on whether or not the maximum generator speed constraint is violated. An initial investigation of the loads on the other turbine components show a reduction in blade and low-speed shaft fatigue and pitch actuation, but a more in-depth loads investigation is left for future work. And note that the gains are also a function of the setting in which it is optimized: These optimized gains reflect the cost function and control goal and environmental setting; we assume that special and fault cases would be handled by a supervisory controller. |

| | |
|---|---|
| Section 3.2.2:

-I like the comparison between the ZOO and the grid search, however, I don't think your example gives the full picture. First, the solution of the grid search depends on the grid itself, and thus, I think in your example the only example which actually makes sense is the example with 10x10 samples. In this view, Figure 6(f) clearly shows that the 6x6 and 8x8 cases are not 'converged' to the real optimal solution. Looking at Figure 6(e) I don't' even think the 10x10 case led to the 'real' optimum. Probably a further refinement of the mesh is required. | **Answer:** We agree. However, the computational burden of grid search methods rapidly increases with the number of search points (following an exponential growth), thus rendering this process computationally infeasible for fine discretizations. This is precisely one issue we address using the proposed ZOO method.

It is also worth pointing out that the proposed approach naturally lends itself to an adaptive tuning strategy in the case of changing operating conditions (where it may not be possible to include an exhaustive search).

**Changes in manuscript:** In several sentences of this section, we clarify that the comparison holds for the pitch control application, e.g.,:

To quantify the performance of the zeroth-order optimization (ZOO) for this pitch control application, we compare…

We acknowledge that, in practice, our grid search optimization would be different:

In practice, we would refine the search area and re-sample based on experience. However, different models may change the re-sampled area and would add a manual step that we can avoid when using the zeroth-order optimization procedure. |
| - Additionally, the effectiveness of the grid search depends on the complexity of the underlying function, so the example you provided gives only a partial conclusion. I would also like to see how the methods compare when tested on (perhaps) 'analytical' functions of growing complexity. | **Answer:** Thank you for this comment. The effectiveness of the grid search depends on whether the grid is sufficiently fine. However, fine grids require a higher computational complexity. For fine grids, the proposed method can provably identify minima (or give points in an interval of a minimum), while requiring fewer functional evaluations. This behavior is irrespective of the functional form of the analytical function.

This is an interesting idea but is outside the scope of this article.

**Changes in manuscript:** We elaborate on the generality of the comparison and explain your idea as a path for future work:

We should note that the comparison presented in this section applies only to this pitch control tuning application. To compare the efficacy of the zeroth-order optimization with a grid search more generally would require comparing functions of different complexities and dimensions, which is outside the scope of this article and we leave for future work. |

| | |
|---|---|
| Section 3.3:
- I think this is the most interesting example, as it deals with a very common problem of control tuning. However, I don't personally like merit functions which are built from a linear combination of requirements, like in this case. What I don't like is that, such a linear combination requires to define the weighting parameters of the various terms in the merit function. In this case, for example, you arbitrarily chose to set k=0.01, and this led to an optimal curve. But, if you change that value, you'll probably obtain a family of curves. So, once again, what is the 'real' optimum? | **Answer:** I agree. A different merit function parameter $\kappa$ would result in a different "optimal" minimum pitch setting, but without more information about the design and constraints of the turbine, we cannot know which $\kappa$ to choose.

**Changes in manuscript:** We note this in Section 3.3:

In future work, a family of optimal minimum pitch control laws, using different values for $\kappa$, could be generated, but would require more global wind turbine design information to determine the design choice. |
| Figure 4:
- Please add (a,b,c,d) identifier in the various subplots.

Sorry for being so verbose, but I enjoyed this paper and I would like to raise some points to further discuss your approach. Thank you for this contribution. | **Answer:** Thank you for the support and productive questions and comments.

**Changes in manuscript:** All figures with more than one subplot have been updated with alphabetic identifiers. |

[revised manuscript text omitted]